# All for One and One for All: A Collaborative FL Framework for Generic Federated Learning with Personalized Plug-ins

## Abstract

Personalized federated learning (PFL) mitigates the notorious data heterogeneity issue in generic federated learning (GFL) by assuming that client models only need to fit on local datasets individually. However, real-world FL clients may meet with test data from other distributions. To endow clients with the ability to handle other datasets, we theoretically formulate a new problem named as Selective FL (SFL), bridging the GFL and PFL together. To practically solve SFL, we design a general effective framework named as *Hot-Pluggable Federated Learning* (HPFL). In HPFL, clients firstly learn a global shared feature extractor. Next, with the frozen feature extractor, multiple personalized *plug-in modules* are individually learned based on the local data and saved in a *modular store* on the server. In inference stage, an accurate selection algorithm allows clients to choose and download suitable plug-in modules from the modular store to achieve the high generalization performance on target data distribution. We conduct comprehensive experiments and ablation studies following common FL settings including four datasets and three neural networks, showing that HPFL significantly outperforms advanced FL algorithms. Additionally, we empirically show the remarkable potential of HPFL to resolve other practical FL problems like continual federated learning and discuss its possible applications in one-shot FL, anarchic FL and an FL plug-in market.

## 1 Introduction

Federated Learning (FL) is an effective framework that lets multiple users or organizations to collaboratively train a machine learning model with data privacy protection. The generic FL (Brendan McMahan et al., 2016) (GFL) was first proposed to obtain a global model (GM) performing well on test data from all clients. However, the performance of the classic FL algorithm FedAvg (Brendan McMahan et al., 2016) suffers from the client drift caused by the data heterogeneity (Kairouz et al., 2019), i.e. different data distributions on clients.

To tackle the data heterogeneity problem, personalized FL (Collins et al., 2021; Chen & Chao, 2021) (PFL) is proposed with assuming that clients only need to perform well on its local test data. Usually, the distribution of local test data is similar to that of its local training data. Thus, PFL usually distinguishes its local models from the GM, and personalizes local models to better adapt to its training data while absorbing knowledge from the global training data. On the local test data, personalized models (PMs) in PFL (PFL-PM) significantly outperform the GM learned by GFL (GFL-GM) (Chen & Chao, 2021).

Table 1: Test accuracy of PMs on generic FL (GFL-PM, G-P) and personalized FL (PFL-PM, P-P) test settings, with ResNet-18 trained on CIFAR-10 dataset.

| Algorithm | FedAvg | | FedPer | | FedRep | | FedRoD | |
|---|---|---|---|---|---|---|---|---|
| Test Settings | G-P | P-P | G-P | P-P | G-P | P-P | G-P | P-P |
| Accuracy | 81.5 | 92.5 | 74.1 | 95.8 | 85.1 | 95.6 | 85.3 | 94.3 |

However, in real-world scenarios, FL users may meet with test data which has different distribution from local training data (Liu et al., 2020; Luo et al., 2019; Hsu et al., 2020), instead similar one ever appeared in other clients . For example, when one is having a trip traveling abroad, weather app on their phone may collect entirely different temperatures from what it used to. Though prediction for the future temperature may be difficult solely with the forecast model trained before, there are users whose model trained on the local temperature which happens to possess similar pattern (if not identical) with the temperature the traveler is trying to predict. We offer more examples showing the

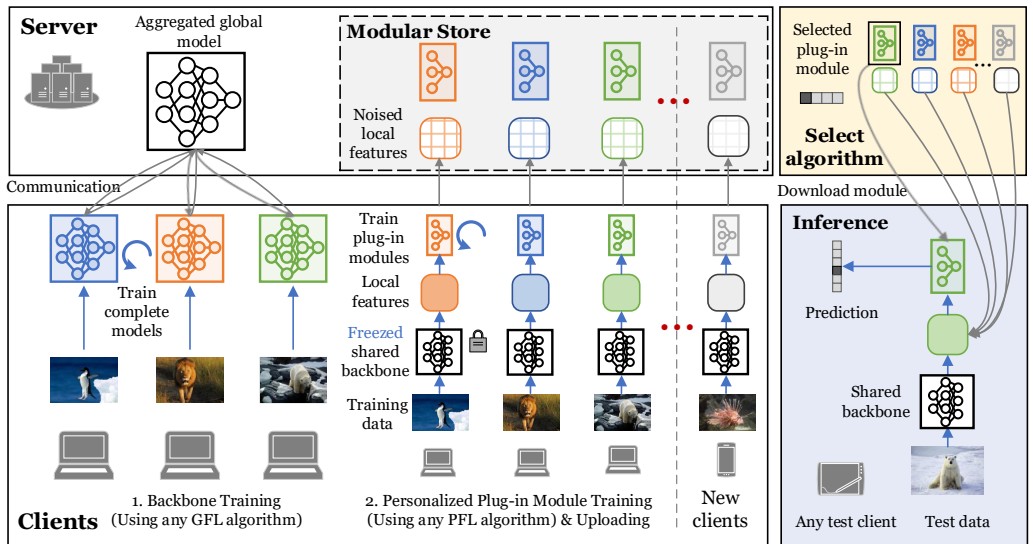

Figure 1: The framework of HPFL. HPFL splits the model into a backbone that learns general representations of all datasets and personalized "plug-in" modules. During inference, clients choose a suitable plug-in module with respect to the test data to complete the inference. New clients in federated continual learning or other scenarios can directly train and contribute new plug-ins without harming the backbone performance.

real scenes of our setting named GFL-PM, in Appendix F.1. In GFL-PM setting, the test set every client encounters comes from local test data of other clients, whose distribution is the same as that of clients' local training data. In these realistic cases, classic PFL algorithms may not be suitable anymore, as the personalized client models cannot generalize well on other test data. We conduct an experiment to illustrate this. We train PMs with advanced PFL algorithms FedPer (Arivazhagan et al., 2019b), FedRep (Collins et al., 2021) and FedRoD (Chen & Chao, 2021), and test their performances on global and local test data. As Table 1 shows, PMs performs well when they are only required to deal with local test data (PFL-PM), but their performances significantly collapse when meeting with global test data (GFL-PM), i.e. clients equally meet with the test data from all clients. This performance degradation of PMs in GFL scenario leads to a practical and fundamental questions:

*Is it possible for FL clients to achieve the generalization performance in GFL as high as PFL?*

To answer this question, we theoretically formulate a new problem called Selective FL (SFL), bridging the GFL and PFL together. Both GFL and PFL can be seen as the special case of the SFL. Its core idea is to let clients select and inference with suitable personalized models (PMs) according to incoming test data. Thus, we give an affirmative answer to the above question. However, the naive solution to SFL faces privacy concerns and large system overheads. To this end, we propose a general effective framework named *Hot-Pluggable Federated Learning* (HPFL) to solve SFL practically.

As shown in Figure 1, HPFL splits the model into two parts: a backbone module (also called feature extractor) and a "plug-in" module. The training process consists of two stages: backbone and plug-in training. When training the backbone, clients exploit GFL algorithms to help them learn a general representation of all datasets. Then, each client individually trains a "plug-in" based on the outputs from the shared backbone with PFL algorithms. All trained "plug-ins" will be uploaded and saved in a "plug-in" store on the server. During inference, clients could download a suitable "plug-in" from the server with respect to the test data, then "plug" it on the backbone to complete the inference.

We summarize our contributions as follows: (1) We identify a substantial gap between GFL and PFL. Then we formulate a new problem SFL to bridge them together to address this performance gap (Section 3); (2) We propose a general efficient and effective framework, HPFL, which practically solves the SFL problem (Section 4); (3) We conduct comprehensive experiments and ablation studies on four datasets and three neural networks to show the effectiveness of HPFL (Section 5); (4) we show the remarkable potential of HPFL in federated continual learning (Section 5.4) and discuss possible applications of HPFL in one-shot FL, anarchic FL and an FL plug-in market (Section 7).

## 2 RELATED WORKS

**Generitic Federated Learning.** The convergence problem of FL with high non-IID data distribution has always been a vital problem in improving the performance of models trained with FL. To resolve this problem, FedProx (Li et al., 2020b) and MOON (Li et al., 2021b) propose to add regularization

terms to mitigate the negative effect caused by data heterogeneity. Some methods modify uploaded gradients to alleviate the dissimilarity (Wang et al., 2020; Karimireddy et al., 2019). Some works share intermediate features (Jeong et al., 2018; Hao et al., 2021) or extra data (Tang et al., 2022) to reduce client drift. Different from these works, we attempt to enhance the GFL performance with personalized models.

**Personalized Federated Learning.** PFL exploits personalizing client models to better suit local heterogeneous training data. Meta-learning (Fallah et al., 2020), knowledge distillation (Yu et al., 2020b; Li & Wang, 2019), adaptive regularization and model mixtures (Hanzely & Richtárik, 2020; Dinh et al., 2020; Deng et al., 2020) are used to enhance personal knowledge learning of models. Some works like LG-FEDAVG (Liang et al., 2020) and LotteryFL (Li et al., 2021a) allow clients to learn different PM structures. FedRep (Collins et al., 2021) and FedRoD (Chen & Chao, 2021) propose to learn a global feature extractor and personalized classifiers. All of these works only consider PMs in PFL settings, i.e. in test time, local PMs only meet test data distribution similar to training distribution. Instead, we excavate the potential of PMs to solve problems in GFL. With divergent purposes, HPFL and those methods train and use these personalized models in quite different way. Unlike those PFL methods, clients can still perform well when meeting unseen test data distribution in HPFL.

**Test-time adaptation & domain adaptation methods in FL.** There exist some works (Peng et al., 2019; Liu et al., 2021) that generalize a federated model trained on multiple source domains to unseen target domains. FedTHE (Jiang & Lin, 2023) discussed test-time distribution shift of PMs, which is similar to our problem setting. These methods enhance federated models by better training schemes. Different from them, HPFL is the first FL framework that selects flexible PMs to achieve this goal, which is orthogonal to existing works. Due to the limited space, we leave a more detailed discussion of the literature review in Appendix A.

## 3 SELECTIVE FL: IMPLEMENTING GENERIC FL FROM PERSONALIZED FL

### 3.1 GENERIC FL

The GFL aims to make $M$ clients collaboratively learn a global model parameterized as $\theta$. Each client has its local data distribution $\mathcal{D}_m$. Thus, the local objective function $\mathcal{L}_m(\theta)$ on client $m$ is also different. The global optimization object of GFL is defined as:

$$\min_{\theta \in \mathbb{R}^d} \mathcal{L}_G(\theta) := \sum_{m=1}^{M} p_m \mathcal{L}_m(\theta) := \sum_{m=1}^{M} p_m \mathbb{E}_{\xi_m \sim \mathcal{D}_m} \ell(f(\theta, \xi_m), \xi_m), \tag{1}$$

where $\xi_m \sim \mathcal{D}_m$ is the data sampled from $\mathcal{D}_m$, $f(\theta, \xi_m)$ is the prediction, $d$ is the number of model parameters, $p_m > 0$ and $\sum_{m=1}^{M} p_m = 1$. Usually, $p_m = \frac{n_m}{N}$, where $n_m$ denotes the number of client $m$'s samples and $N = \sum_{m=1}^{M} n_m$. *GM* refers to the model obtained from optimizing GFL.

### 3.2 PERSONALIZED FL

Different from the object function of GFL, the PFL aims to learn multiple personalized models which fit well on different datasets individually: (Li & Wang, 2019; Chen & Chao, 2021; Li et al., 2021c):

$$\min_{\Omega, \theta_1, ..., \theta_M} \mathcal{L}_P(\Omega, \theta_1, ..., \theta_M) := \sum_{m=1}^{M} p_m \mathbb{E}_{\xi_m \sim \mathcal{D}_m} \ell(f(\theta_m, \xi_m), \xi_m) + \mathcal{R}(\Omega, \theta_1, ..., \theta_M), \tag{2}$$

where $\mathcal{R}$ is a regularizer (Chen & Chao, 2021) that varies with different algorithms, $\Omega$ is used to collaborate clients. We call each obtained locally personalized model $\theta_m$ as *PM*.

### 3.3 WHEN PM MEETS GFL

In practice, PMs of clients may meet test data from other clients. Therefore, the learned PMs $\theta_1, ..., \theta_M$ need to perform well on all local data $\mathcal{D}_1, ..., \mathcal{D}_M$. We formulate the corresponding optimization goal with PMs in GFL scenario (GFL-PM) is:

$$\min_{\Omega, \theta_1, ..., \theta_M} \mathcal{L}_{P-G}(\Omega, \theta_1, ..., \theta_M) = \frac{1}{M} \sum_{i=1}^{M} \sum_{m=1}^{M} p_m \mathbb{E}_{\xi_m \sim \mathcal{D}_m} \ell(f(\theta_i, \xi_m), \xi_m) + \mathcal{R}(\Omega, \theta_1, ..., \theta_M), \tag{3}$$

which can be seen as a combination of GFL (Eq. 1) and PFL (Eq. 2): each PM is optimized to minimize the $\ell$ on all $\mathcal{D}_m$, $m \in 1, ..., M$. When not personalize $\theta_i$ on $\mathcal{D}_i$, Eq. 3 is reduced to GFL . And if each client's PM only needs to perform well on its local data, Eq. 3 turns into PFL.

One may think that there is no need to endow PMs with global generalization performance because one can optimize GFL to obtain a GM that generalizes well on all local datasets $\{\mathcal{D}_m, m \in \{1, ..., M\}\}$. However, theoretically and empirically, optimization of GM is difficult (Karimireddy et al., 2019; Woodworth et al., 2020) under communication cost and data heterogeneity constraints. Additionally, PMs' performance on local test data (PM on PFL) is usually significantly better than that of GM on global test data (GM on GFL) (Chen & Chao, 2021; Collins et al., 2021).

However, PMs after PFL usually cannot achieve better performance on unseen data distributions than GM in GFL (Chen & Chao, 2021). FedRoD (Chen & Chao, 2021) simultaneously optimizes $\mathcal{L}_G$ and $\mathcal{L}_P$, aiming to learn models that perform well both in GFL and PFL. This shares a similar spirit of optimizing GFL-PM problem (Eq. 3). However, PMs obtained from FedRoD remain a trade-off between minimizers of PFL and GFL. It is challenging to obtain model parameters that are both minimizers of GFL and PFL simultaneously. Next, we show that GFL-PM can be naturally transformed into a Selective FL (SFL) problem (Eq. 5), which involves optimizing PFL and a model selection problem (Eq. 6 in section 3.4). And the solution of SFL could serve as the minimizer of both GFL and PFL.

## 3.4 SELECTIVE FL

Successful personalization on client $i$ means the following equation (Chen & Chao, 2021; Kairouz et al., 2019; Tan et al., 2022a).

$$\mathbb{E}_{\xi_m \sim \mathcal{D}_m}\ell(f(\theta_i, \xi_m), \xi_m) \geq \mathbb{E}_{\xi_m \sim \mathcal{D}_m}\ell(f(\theta_m, \xi_m), \xi_m), i \neq m, \tag{4}$$

which means that for any client $i$, its PM outperforms than all PMs of other clients (Chen & Chao, 2021). Now, we are ready to state the following theorem (proof in Appendix B.1).

**Theorem 3.1.** *With Equation 4 and the PMs obtained from optimizing Equation 2 as:* $\Omega^{pfl}, \theta_1^{pfl}, ..., \theta_M^{pfl} = \arg\min_{\Omega, \theta_1, ..., \theta_M} \mathcal{L}_P(\Omega, \theta_1, ..., \theta_M)$, *we have*

$$\mathcal{L}_{P-G}(\Omega, \theta_1, ..., \theta_M) \geq \mathcal{L}_P(\Omega^{pfl}, \theta_1^{pfl}, ..., \theta_M^{pfl}).$$

*Remark* 3.1. Theorem 3.1 implies that $\mathcal{L}_{P-G}$ is lower bounded by the minimum of $\mathcal{L}_P$.

Theorem 3.1 inspires us to think about a question: *Is it possible to exploit PMs to improve the generalization performance on the global dataset?* Based on Equation 4, the intuitive solution is to design a new forward function $\widehat{f}$ to make client $i$ generate the same outputs of $f(\theta_m^{pfl}, \xi_m)$ when meeting data $\xi_m \sim \mathcal{D}_m$. Thus, we propose the Selective FL (SFL) problem as the following:

$$\min_{\mathcal{H}} \mathcal{L}_S(\Theta, \mathcal{H}) := \sum_{m=1}^M p_m \mathbb{E}_{\xi_m \sim \mathcal{D}_m}\ell(\widehat{f}(\Theta, \xi_m, \mathcal{H}), \xi_m) \tag{5}$$

$$s.t. \ \widehat{f}(\Theta, \xi_m, \mathcal{H}) = f(\theta_s^{pfl}, \xi_m), \ s = S(\Theta, \xi_m, \mathcal{H}) \tag{6}$$

where $\Theta = \{\Omega^{pfl}, \theta_1^{pfl}, ..., \theta_M^{pfl}\} = \arg\min_{\Omega, \theta_1, ..., \theta_M} \mathcal{L}_P(\Omega, \theta_1, ..., \theta_M)$, $S$ is called selection function that outputs the model index to select (or say "generate") a model from the PMs based on the input $\xi_m$ and the auxiliary information $\mathcal{H}$ (We will illustrate what can be the auxiliary information in Section 4). Now, we can state the following theorem to illustrate that we can solve problem 3 by SFL (proof in Appendix B.2):

**Theorem 3.2.** *With equation 4,* $\Omega^{pfl}, \theta_1^{pfl}, ..., \theta_M^{pfl} = \arg\min_{\Omega, \theta_1, ..., \theta_M} \mathcal{L}_P(\Omega, \theta_1, ..., \theta_M)$ *and the* $\mathcal{H}^*$ *that guarantees* $\theta_m^{pfl} = s(\Theta, \xi_m, \mathcal{H})$, *we have*

$$\mathcal{L}_{P-G}(\Omega, \theta_1, ..., \theta_M) \geq \mathcal{L}_P(\Theta) = \mathcal{L}_S(\Theta, \mathcal{H}^*).$$

*Remark* 3.2. Theorem 3.2 shows that if we can accurately select $\theta_m^{pfl}$ out from all PMs when meeting data samples $\xi_m \sim \mathcal{D}_m$, the solution of SFL is also the lower bound of GFL-PM (Eq. 3). Therefore, solving SFL means that clients can achieve a generalization performance in GFL as high as PFL.

## 4 HPFL: A GENERAL EFFECTIVE FRAMEWORK TO SOLVE SELECTIVE FL

In this section, we will first illustrate that directly selecting PM faces some fatal obstacles, including the large system overheads and privacy concerns in Section 4.1. Then, we introduce the design of HPFL in Section 4.2 with the Algorithm 1. Lastly, the selection method is introduced in Section 4.3.

## 4.1 PROBLEMS OF DIRECTLY SELECTING PM

With PMs $\Theta = \{\Omega^{pfl}, \theta_1^{pfl}, ..., \theta_M^{pfl}\} = \arg\min_{\Omega, \theta_1, ..., \theta_M} \mathcal{L}_P(\Omega, \theta_1, ..., \theta_M)$, an intuitive idea is to choose PM $i$ based on the similarity between its local data $\mathcal{D}_i$ and the input data $\xi_m \sim \mathcal{D}_m$, thus the selection function 6 is implemented as: $s = S_\xi(\Theta, \xi_m, \mathcal{H}) = \arg\min_{i \in \mathcal{M}} d(\mathcal{D}_i, \xi_m)$, where $d(\cdot, \cdot)$ is any distance measure, then do inference as $f(\theta_s^{pfl}, \xi_m)$. However, accessing data of other clients will cause **privacy concerns**. Moreover, communicating the whole model parameter $\theta_m$ is impractical due to **large system overhead**, especially for large language models and many clients.

## 4.2 DESIGN OF HPFL

**Training the complete model $\theta$.** First, with any GFL algorithm, HPFL obtains a model $\theta$ that performs well (not as good as PMs in PFL) on all client datasets. Thus, the model $\theta$ owns a backbone $g$ that can extract general features from all client datasets. Due to the limited space, we chose the classic GFL algorithm FedAvg (McMahan et al., 2017) in our experiments. Future works can explore other advanced GFL algorithms to learn a better $\theta$.

**Training personalized plug-in module $\theta_m^\rho$.** Usually, after training, early layers of a model learn more general features than late layers (Yosinski et al., 2014; Asano et al., 2020), which means that early layers can extract useful

---

**Algorithm 1** HPFL.

**Initialization:** server distributes the initial model $\theta^0$ to all clients.
**1. Training the complete model $\theta$:**
**for** each round $r = 0, 1, \cdots, R$ **do**
    server samples a set of clients $\mathcal{S}_r \subseteq \{1, ..., M\}$.
    server communicates $\theta^r$ to clients $m \in \mathcal{S}_r$.
    **for** each client $m \in \mathcal{S}^r$ **in parallel do**
        $\mathcal{C}_m^{r+1} \leftarrow \text{LocalTraining}(\mathcal{D}_m, \theta^r)$ (GFL) .
    **end for**
    $\theta^{r+1} \leftarrow \text{ServerUpdate}(\mathcal{C}_m^{r+1} | m \in \mathcal{S}^r)$ (GFL).
**end for**

**2. Training personalized plug-in module $\theta_m^\rho$:**
**for** each client $m \in \mathcal{M}$ **in parallel do do**
    Clients share and freeze the $\theta^g$,
    Clients design personalized $\theta_m^\rho$.
    Training $\theta_m^\rho$ with object function 7 (PFL).
    Obtaining auxiliary information $\mathcal{H}_m$ (e.g. noised features explained in Section 4.3 in detail.)
    for **plug-in selection**.
    Upload $\theta_m^\rho$ and $\mathcal{H}_m$ to server.
**end for**
Server stores $\theta_m^\rho$ and $\mathcal{H}_m$.

**HPFL Inference($\theta^g, \mathcal{D}_{test}$):**
$i \leftarrow \text{SelectPlugIn}(\mathcal{D}_{test}, \theta^g, \mathcal{H})$.
Get output $\leftarrow \rho_i \circ g(\xi | \xi \sim \mathcal{D}_{test})$.

---

features from more datasets than late layers, but late layers are more specific to some particular datasets. Inspired by this, HPFL decomposes the model as $f_m = \rho \circ g$ for each client $m$. As shown in Figure 1, $g$ is a feature extractor, and $\rho$ is a model head that outputs the final model prediction.

Clients can design a new personal plug-in module $\rho_m$ (or say model head) different from the original head $\rho$, based on different computation characteristics. Then, with the frozen general feature extractor $g$, each client individually trains personalized $\rho_m$ on local datasets $\mathcal{D}_m$ by optimizing:

$$\min_{\theta_m^\rho} \mathcal{L}_P(\theta_m) := \mathbb{E}_{\xi_m \sim \mathcal{D}_m} \ell(\rho_m \circ g(\xi_m), \xi_m). \tag{7}$$

Now, each client obtains a PM $f_m = \rho_m \circ g$, which enhances the generalization performance of $\rho_m \circ g$ on $\mathcal{D}_m$, which is usually better than original GM $f = \rho \circ g$ due to the personalization. Thus, the $\theta_m^{pfl}$ in SFL problem 5 can be constructed by $\theta^g$ and $\theta_m^\rho$, inference becomes as $f(\theta_m^{pfl}, \xi_m) = \rho_m \circ g(\xi_m)$.

**Inference and selecting plug-in module.** In HPFL, we define some auxiliary information $\mathcal{H}_m$ that will be exploited to select plug-in module and propose specific forms of it in Section 4.3. When training $\theta_m^\rho$, $\mathcal{H}_m$ are collected by clients and uploaded to the server. Note that as a general framework, HPFL does not limit the specific form of $\mathcal{H}_m$, which depends on the selection method. In this paper, We introduce a distance-based selection method in Section 4.3. We discuss and analyze the potential privacy risk of sharing the plug-ins in Appendix E.1.

## 4.3 SELECTION METHODS

Decomposing the DL model also helps to avoid accessing the raw data $\xi_m \sim \mathcal{D}_m$. With the help of the shared feature extractor $g$, we can select the $\rho_m$ based on the intermediate features $h_m = g(\xi_m)$ rather than $\xi_m$ itself. There have been some works that exploit sharing intermediate features to improve FL (He et al., 2020a; Lin et al., 2020; Luo et al., 2021; Liang et al., 2020).

**Distance based methods.** Intuitively, now that each $\rho_m$ is trained based on local features $h_m$, we only need to compare the similarity between $h_m$ and $h_{test} = g(\xi_{test})$, where $\xi_{test}$ is the data that

needs testing. Now, the select problem turns from equation 6 into:

$$S_{dist}(d, h_{test}, \hat{h}_1, ..., \hat{h}_M) = \arg\min_{m \in \mathcal{M}} d(\hat{h}_m, h_{test}), \qquad (8)$$

in which $\hat{h}_m = (h_m + \kappa * \epsilon)/(1 + \kappa)$, where $\epsilon \sim \mathcal{N}(\mu_m, \sigma_m)$ is the noise to enhance privacy protection. The $\mu_m$ and $\sigma_m$ are mean and variance of features $h_m$. $\kappa$ is the coefficient controlling the relative magnitude between Gaussian noise and the features. Clients receive the noised features $\hat{h}_m$ for plug-in selection. In this selection method, the $\mathcal{H}_m = \hat{h}_m$. We discuss the potential privacy risk of the selection method in Appendix E.2, and testify sharing the noised feature stays safe from model inversion attack. MMD measures the Hilbert-Schmidt norm between kernel mean embedding of empirical joint distributions of source and target data (Long et al., 2017). In HPFL, we utilize it to measure the distance between features that plug-in modules train on and features of test data. Note that we can also choose other distance measures. Due to page limitation, results of HPFL based on SVCCA (Raghu et al., 2017) and CKA (Kornblith et al., 2019) are shown in Appendix D. We also provide an out-of-distribution confidence based selection method and its results in Appendix D.2.

## 5 EXPERIMENTS

### 5.1 EXPERIMENT SETUP

**Federated Datasets and Models.** We conduct experiments on four commonly used image classification datasets in FL, including CIFAR-10 (Krizhevsky et al., 2009), CIFAR-100 (Krizhevsky et al., 2009), Fashion-MNIST (Xiao et al., 2017), and Tiny-ImageNet (Le & Yang, 2015), with Latent Dirichlet Sampling (Dir) partition method ($\alpha = 0.1$ and $0.05$.) to simulate the data heterogeneity following (He et al., 2020b; Li et al., 2021b; Luo et al., 2021). We also evaluate the scalability of our proposed methods with different number of clients ($M$ =10 and 100). We train ResNet-18 (He et al., 2016), MobileNet and a simple-CNN on all datasets. We run all for 1000 communication rounds, with 1 local epoch in each round. Hyper-parameters and more details are explained in Appendix C.

**Baselines and Metrics.** We compare HPFL with classic GFL algorithm FedAvg (McMahan et al., 2017), advanced PFL algorithms including FedPer (Arivazhagan et al., 2019a), FedRep (Collins et al., 2021), PerFedMask (Setayesh et al., 2023), FedRoD (Chen & Chao, 2021) which is for both GFL and PFL, and a test-time adaption method FedTHE (Jiang & Lin, 2023). For all algorithms, we validate the learned global model (GM) on the global test dataset (GFL), and the personalized models (PM) on the personalized dataset (PFL), also PMs on GFL. Note that PFL only focuses on individually testing on local datasets instead of all datasets. More details about metrics are stated in Appendix C.

### 5.2 EXPERIMENT RESULTS

**HPFL consistently outperforms baselines in PM on GFL while comparable with classic PFL methods in classic personalized setting.** As shown in Table 2, in **GFL-PM** setting, HPFL performs the best in all of methods and most by a large margin, even surpasses accuracies in GFL-GM in most cases, while baselines perform poorly due to a lack of adaption to the test data. We attribute the significant performance gain to adaptation to test data implemented with precise plug-in selection, which we are going to discuss in Section 5.3. It is worth noting that FedTHE also attempts to adapt its model using test data, but only with the ensemble of its locally personalized and global classifier, thus not fully utilizes the knowledge of other clients and performs worse than HPFL. In terms of **GFL-GM accuracy**, HPFL actually shares the same GM with GFL backbone training method (in our case, i.e. FedAvg), so its GFL-GM accuracy is exactly the same as that of FedAvg and outperforms the classic PFL algorithms only focusing on PFL performance like FedPer (Arivazhagan et al., 2019a). As for **PFL-PM accuracy**, our proposed method HPFL reports comparable results to the PFL baselines.

**HPFL maintains fairly excellent robustness against non-IID degree.** As shown in Table 2, the accuracy of HPFL is not only highest in GFL-PM, but also increases when the heterogeneity increases from $Dir(0.1)$ to $Dir(0.05)$ in a similar way as in PFL-PM in some cases. From this phenomenon, we infer that HPFL exploits local information from clients to ensemble a model in the form of plug-ins. The server holds these local information in the form of plug-ins instead of fusing these local knowledge in a single model, thus prevents the original local information from being corrupted in model aggregation as it occurs in highly heterogeneous data, and maintains a robustness against non-IID, which is a common issue in Federated Learning.

Table 2: Experiment results. Noisy coefficient $\kappa$=1. §: we focus more on GFL setting. Numbers in **ForestGreen** highlight highest values in GFL setting. *: FedAvg fine-tunes the whole model instead of partial model as in HPFL. Plug-in selection is implemented with MMD. $E_p$ denotes the epoch of fine-tuning.

| Clients | 10 (sample 50% each round) | | | | | | 100 (5% each round) | | | | | |
|---|---|---|---|---|---|---|---|---|---|---|---|---|
| Non-IID | Dir(0.1) | | | Dir(0.05) | | | Dir(0.1) | | | Dir(0.05) | | |
| Test Set | GFL§ | | PFL | GFL§ | | PFL | GFL§ | | PFL | GFL§ | | PFL |
| Method/Model | GM | PM | PM | GM | PM | PM | GM | PM | PM | GM | PM | PM |
| **CIFAR-10** | | | | | | | | | | | | |
| FedAvg $E_p=1$* | 81.5 | - | 92.5 | 62.4 | - | 96.1 | 73.6 | - | 90.9 | 47.9 | - | 91.5 |
| FedAvg $E_p=10$* | 81.5 | - | 92.8 | 62.4 | - | 92.7 | 73.6 | - | 91.6 | 47.9 | - | 93.4 |
| FedPer | 74.1 | 40.9 | **95.8** | 58.7 | 27.3 | 96.4 | 44.5 | 20.6 | 89.7 | 24.0 | 14.3 | 89.9 |
| FedRoD | 85.3 | 41.6 | 94.3 | 67.6 | 26.8 | **96.9** | 74.0 | 20.1 | 87.4 | 66.7 | 15.6 | 91.2 |
| FedRep | 85.1 | 51.3 | 95.6 | 73.2 | 30.2 | 85.3 | 66.5 | 27.4 | 89.3 | 59.2 | 20.4 | 89.1 |
| PerFedMask $E_p=5$ | 57.8 | 23.4 | 83.1 | 31.8 | 15.1 | 83.1 | 53.8 | 15.6 | 82.1 | 35.0 | 12.5 | 87.6 |
| FedTHE | 86.4 | 51.6 | 90.6 | 68.0 | 32.6 | 89.2 | 74.0 | 41.5 | 88.3 | 66.7 | 43.3 | 87.9 |
| HPFL $E_p=1$ | 81.5 | 95.4 | 95.4 | 62.4 | 96.0 | 96.0 | 73.6 | **88.6** | 94.9 | 47.9 | **82.2** | 93.9 |
| HPFL $E_p=10$ | 81.5 | **95.7** | 95.7 | 62.4 | **96.3** | 96.3 | 73.6 | 85.7 | **95.7** | 47.9 | 81.8 | **95.3** |
| **FMNIST** | | | | | | | | | | | | |
| FedAvg $E_p=1$* | 86.0 | - | 98.0 | 76.1 | - | 99.1 | 90.2 | - | 97.2 | 86.1 | - | 97.9 |
| FedAvg $E_p=10$* | 86.0 | - | 98.2 | 76.1 | - | 99.1 | 90.2 | - | 97.8 | 86.1 | - | 98.4 |
| FedPer | 73.5 | 39.0 | 87.5 | 64.1 | 27.5 | 99.1 | 69.0 | 29.1 | 95.9 | 44.8 | 22.6 | 96.8 |
| FedRoD | 87.4 | 44.1 | 98.1 | 72.5 | 29.3 | 98.9 | 88.9 | 47.0 | 98.5 | 84.8 | 35.3 | 98.2 |
| FedRep | 87.0 | 43.0 | 97.5 | 74.7 | 39.5 | 98.0 | 88.2 | 72.4 | 97.9 | 84.4 | 59.6 | 98.3 |
| PerFedMask $E_p=5$ | 80.1 | 30.8 | 95.8 | 47.6 | 27.1 | 96.9 | 89.3 | 23.0 | 93.5 | 91.9 | 21.3 | 96.5 |
| FedTHE | 87.3 | 64.8 | 94.6 | 73.6 | 59.0 | 97.7 | 88.6 | 17.1 | 93.4 | 84.8 | 74.7 | 95.7 |
| HPFL(MMD) $E_p=1$ | 86.0 | 98.3 | 98.3 | 76.1 | 99.0 | 99.1 | 90.2 | 97.6 | 97.9 | 86.1 | 81.4 | 98.1 |
| HPFL(MMD) $E_p=10$ | 86.0 | **98.4** | 98.4 | 76.1 | **99.1** | 99.2 | 90.2 | **97.9** | 98.8 | 86.1 | 74.1 | **98.7** |
| **CIFAR-100** | | | | | | | | | | | | |
| FedAvg $E_p=1$* | 69.1 | - | 79.5 | 65.3 | - | 77.4 | 59.7 | - | 60.0 | 47.9 | - | 69.2 |
| FedAvg $E_p=10$* | 69.1 | - | 72.3 | 65.3 | - | 80.9 | 59.7 | - | 66.7 | 47.9 | - | 75.1 |
| FedPer | 38.6 | 22.5 | 74.6 | 33.9 | 17.8 | 82.8 | 13.2 | 7.0 | 49.1 | 4.1 | 2.7 | 46.7 |
| FedRoD | 69.4 | 32.5 | 77.2 | 67.0 | 23.6 | 78.5 | 52.8 | 11.2 | 55.4 | 48.4 | 7.3 | 66.3 |
| FedRep | 68.4 | 42.6 | 72.4 | 65.0 | 37.3 | 81.2 | 47.9 | 18.6 | 56.5 | 43.3 | 14.1 | 65.3 |
| PerFedMask $E_p=5$ | 47.3 | 7.0 | 40.0 | 49.4 | 7.0 | 39.7 | 41.7 | 3.8 | 35.8 | 42.1 | 3.6 | 35.2 |
| FedTHE | 69.8 | 20.5 | 69.0 | 66.9 | 14.2 | 73.2 | 53.7 | 7.9 | 51.9 | 48.4 | 3.6 | 60.9 |
| HPFL(MMD) $E_p=1$ | 68.6 | **74.8** | 83.3 | 65.3 | **75.8** | 87.4 | 59.7 | **63.8** | 81.2 | 47.9 | **72.3** | 84.1 |
| HPFL(MMD) $E_p=10$ | 68.6 | 72.2 | **85.7** | 65.3 | 73.9 | **88.8** | 59.7 | 55.7 | **84.1** | 47.9 | 70.9 | **86.4** |
| **Tiny-ImageNet-200** | | | | | | | | | | | | |
| FedAvg $E_p=1$* | 56.5 | - | 69.5 | 54.9 | - | 75.3 | 47.2 | - | 53.3 | 42.1 | - | 58.0 |
| FedAvg $E_p=10$* | 56.5 | - | 66.8 | 54.9 | - | 73.6 | 47.2 | - | 67.5 | 42.1 | - | 68.9 |
| FedPer | 16.3 | 0.5 | 0.5 | 13.4 | 0.5 | 0.5 | 2.4 | 1.8 | 23.5 | 1.3 | 25.1 | 1.0 |
| FedRoD | 57.5 | 26.1 | 68.5 | 55.3 | 12.9 | 52.9 | 48.6 | 49.3 | 9.6 | 43.7 | 5.9 | 53.7 |
| FedRep | 56.1 | 28.7 | 55.4 | 54.5 | 31.8 | 69.6 | 46.4 | 18.6 | 52.5 | 40.3 | 12.8 | 58.6 |
| PerFedMask $E_p=5$ | 26.9 | 6.6 | 35.9 | 23.2 | 4.2 | 31.3 | 29.9 | 1.9 | 23.5 | 18.7 | 1.6 | 32.6 |
| FedTHE | 57.5 | 15.6 | 60.4 | 55.3 | 14.1 | 71.2 | 48.6 | 15.8 | 55.9 | 43.7 | 10.3 | 56.9 |
| HPFL(MMD) $E_p=1$ | 56.5 | **51.9** | 70.8 | 54.9 | 58.5 | 74.7 | 47.2 | **50.7** | 71.3 | 42.1 | **47.1** | 74.7 |
| HPFL(MMD) $E_p=10$ | 56.5 | 50.9 | **73.7** | 54.9 | **58.8** | **77.0** | 47.2 | 48.0 | **73.2** | 42.1 | 43.9 | **76.5** |

**HPFL has excellent scalability in terms of performance in accuracy.** HPFL adopts a one-client-one-plug method to better modify final inference models according to the data distribution of clients' local data. In this way, HPFL has inherent ability to allow more clients to come and go freely in the FL system. From Table 2, we observe that other PFL methods met with extreme problems when dealing with the situation that the number of clients was larger ($\mathcal{M}$=100), with most of the accuracies lower than 30% on CIFAR-10, 20% on CIFAR-100. However, though with a little decay in accuracy, HPFL is still applicable in the situation where the system included larger number of clients.

Table 3: Results with different architectures.

| Architecture | Mobilenet | | | Simple-CNN | | |
|---|---|---|---|---|---|---|
| Method/Model | GM | PM | PM | GM | PM | PM |
| FedAvg | 55.7 | - | 92.3 | 64.6 | - | 85.4 |
| FedPer | 53.7 | 10.0 | 10.0 | 44.1 | 27.6 | 85.5 |
| FedRoD | 76.3 | 36.1 | 92.3 | 67.1 | 28.8 | 83.5 |
| FedRep | 74.1 | 35.8 | 85.0 | 54.6 | 10.0 | 10.0 |
| PerFedMask | 13.0 | 19.0 | 76.4 | 31.5 | 10.0 | 50.5 |
| FedTHE | 76.3 | 45.4 | 82.7 | 67.1 | 45.1 | 70.9 |
| HPFL | 55.7 | **92.8** | 92.8 | 64.6 | **87.8** | 87.8 |

**A generalized framework applicable to different model architecture.** As a general FL framework, HPFL can be seamlessly applied to model architectures where parameter decoupling is available. We deploy it on three different model architectures (ResNet-18, MobileNet (Howard et al., 2017), and a simple-CNN structure whose architecture is the same as simple-CNN in (Tang et al., 2022)), and HPFL outperforms baselines we use in the main experiment with all of the architectures, showing that HPFL can be extensively employed in different FL systems and improve their performance of GFL and adaptation ability to new clients. Results are in Table 3. Moreover, HPFL can exploit backbones trained with all kinds of GFL algorithms. An ablation study on GFL methods used to learn feature extractor of HPFL is demonstrated in Appendix D.6.

**A win-win deal: Efforts to protect privacy is not contradictory to the performance of HPFL.** In HPFL, clients share auxiliary information with the server, which may raise privacy concern. To protect clients from the risk of data breaches during communication or improper storage on the server, we add noise to the auxiliary information. However, we surprisingly found that noise will not damage the performance of HPFL as shown in Table 4. We attribute the robustness toward noise to robust selection method of HPFL, which we study later in Section 5.3. Results of the model inversion attack against HPFL are shown in Appendix E.

Table 4: Accuracy of Different $\kappa$

| $\kappa$ | 0 | 1 | 10 | 100 | 1000 |
|---|---|---|---|---|---|
| Accuracy | 95.4 | 95.4 | 95.4 | 95.4 | 95.4 |

**The more flexible the models are, the better?** As shown in Figure 2, the accuracy of HPFL continuously decreases with the increasing number of plug-in layers, we propose two possible reasons leading to the phenomenon: (1) local clients' samples are not sufficient for training big-scale plugs, resulting severe overfitting issue, and (2) The selection methods may not be suitable for middle features. However, according to Table 2, we believe that fine-tuning larger plug-ins does not lead to such a performance degradation, because FedAvg fine-tunes on the whole model without significant performance loss. Therefore, it is natural to give attention to the potential trouble big plug-ins

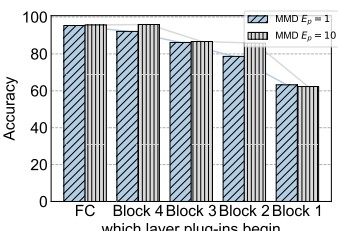

Figure 2: Different plug-in layers.

may cause in plug-in selection. In Section 5.3, we conduct experiments to testify the speculation that the performance loss when increasing the plug-in layer is mainly due to the degradation of plug-in selection. Due to the page limit, we aim to provide an intuitive explanation in Appendix D.1.

## 5.3 SELECTION ACCURACY

Plug-in selection plays an important role in HPFL, so here we study how it is affected by the magnitude of noise added on features and the number of plug-in layers. Experiments in this section are carried out with $\alpha$=0.1, $\mathcal{M}$=10 on CIFAR-10 dataset, we include the results of additional configurations in Appendix D.3.

We observed the expected phenomenon conforming to our conjecture in Section 5.2 that it is harder for selection methods to correctly select plug-ins with more layers. With the increasing number of plug-in layers, the score map gradually begins to change. However, until it actually start to influence the result of selection, the performance of HPFL gets unaffected.

Observed from Figure 3, despite the slight variation in the heatmaps of MMD score with the noise coefficient, selecting plug-in with the lowest MMD score instead of combining plug-ins with MMD score adds robustness towards noise to HPFL. The accuracy shows in Table 4.

## 5.4 FEDERATED CONTINUAL LEARNING

Federated continual learning (FCL) (Yoon et al., 2021) is a new problem where clients join FL training after initial training. The trained model must retain previous dataset knowledge and perform well on data from newly arrived clients. HPFL can address the forgetting problem of FCL by preserving previous training knowledge in a personalized plug-in and providing it for client inference as shown in Table 5. It is an application of HPFL on the temporal scale, where clients collaboratively learn models that generalize well over time.

Table 5: Results of FCL

| Test data | GFL | |
|---|---|---|
| Method/Model | GM | PM |
| Naive FCL | 69.5 | 58.4 |
| FCL under HPFL | 62.2 | **80.9** |

We present more details about the experiment and discussion in Appendix D.5.

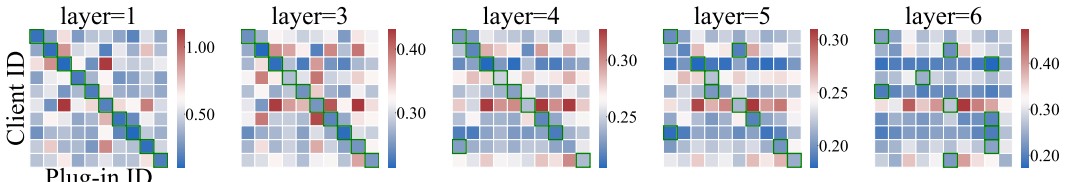

Figure 3: Selection score maps with different noise coefficient. Blocks with green anchor mean the corresponding client selects the plug-in and download it. Blocks with green anchor lying in diagonal indicate that clients choose plug-ins of themselves when met with their own test data, which conforms to the aim of selection methods.

Figure 4: Selection score maps with different number of plug-in layers

## 6 LIMITATIONS

**Accurate Plug-in Selection.** As an initial trial, our proposed plug-in selection methods select sub-optimal plug-ins in some circumstances as shown in Figure 4, 3, 12 and 9 etc.. Future works may consider to design more accurately and robust selection methods.

**Training The Feature Extractor.** In this work, we only consider using the classic GFL algorithm FedAvg to train the feature extractor while achieving superior performance. Designing methods to obtain a better feature extractor will be an important direction to enhance the practicality of HPFL.

## 7 BROADER IMPACT

**Federated continual learning.** As shown in Section 5.4, HPFL can effectively tackle the forgetting problem in FCL, benefit from its ability to losslessly maintain the knowledge learned in a dataset and recover it when in need. The superiority of HPFL meets the need of FCL: FCL can be regarded as a distribution shift problem at Federated Learning within the temporal scale since the distribution of training data shifts as participants of FL change with time.

**One-shot FL.** Once an average backbone is accessible like a pre-trained model, HPFL is able to directly train plug-ins in a single communication round and go straight into the inference stage. The same procedure also applies to the situation where a new client takes part in the FL system.

**Anarchic FL.** In anarchic FL (Yang et al., 2022), clients can decide to join or quit training at any time, which severely harms FL convergence. To this end, HPFL naturally allows this kind of working paradigm. Like one-shot FL, once the backbone is accessible, any aggregation operation is not in demand for HPFL, so the server does not rely on timely responses of clients and will not be disturbed by stale model updates. Clients can finish training and uploading plug-ins at any time.

**FL plug-in market.** HPFL provides the possibility of constructing a more free and transparent model market, and customers can have better confidence knowing the plug-in they are purchasing is able to meet their requirements with a fair plug-in selection mechanism. Plug-in providers can obtain commercial benefits from this market.

## 8 CONCLUSION

In this paper, we explore how to improve the generalization performance when PMs meet test data from other clients. We formalize the SFL to bridge the GFL and PFL together. Then, We propose HPFL to practically solve the SFL. We verify the effectiveness and robustness of HPFL through comprehensive experiments. And we further experimentally verify the remarkable potential of HPFL to resolve other practical FL problems like FCL. Future work can consider to explore new plug-in selection methods, or applying HPFL into more FL related problems.

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

APPENDIX

## A MORE RELATED WORK

### A.1 GENERIC FEDERATED LEARNING

The convergence problem of FL with high non-IID data distribution has always been an important problem in improving the performance of models trained with FL. To resolve this problem, Fed-Prox (Li et al., 2020b) and MOON  (Li et al., 2021b) propose to add new model regularization terms to mitigate the client drift caused by data heterogeneity. There are also some methods modifying the uploaded gradient to alleviate the dissimilarity of gradients (Wang et al., 2020; Karimireddy et al., 2019). With a level of privacy protection, some works propose to share intermediate features (Jeong et al., 2018; Hao et al., 2021) or extra data (Tang et al., 2022; Shin et al., 2020; Lin et al., 2020) to reduce the gradient variance.

### A.2 PERSONALIZED FEDERATED LEARNING

Different from the GFL methods that aim to directly reduce the gradient dissimilarity, PFL exploits the heterogeneous data to personalize client models to better suit the local training data.

Recently, several works have proposed to apply Model-Agnostic Meta-Learning (Finn et al., 2017) to Federated Learning for faster adaptation on local training data in clients. The Model-Agnostic Meta-Learning (Finn et al., 2017) (MAML) aims to meta-learn a global model, which will be broadcasted to different users to learn a local model adapted to different datasets. Per-FedAvg (Fallah et al., 2020) makes use of MAML to learn personalized models more efficiently. It first finds an initial global shared model with second-order gradient information, and then the global model is fine-tuned by local models with only several iterations to suit the local datasets.

Knowledge distillation is also used to promote efficient local adaptation of personalized models (Yu et al., 2020b). Specifically, a federated teacher model $G_T$ and an adapted student model $G_S$ are defined with the same structure. $G_S$ is initialized with $G_T$, which has been trained through a common dataset shared across clients. And $G_S$ is trained by local private datasets. However, in this method, the global model $G_T$ won't get optimized as time goes on. It is more like a local fine-tuning technology rather than federated learning. Some works (Hanzely & Richtárik, 2020; Dinh et al., 2020; Deng et al., 2020) utilize some regularization and adaptive model mixture to learn personalized models. FedMD (Li & Wang, 2019) proposes a federated learning framework based on knowledge distillation using a shared dataset, on which clients transfer knowledge through mimicking the outputs of other client models. With knowledge distillation, it allows clients to independently design their own model architectures with their local private datasets.

In addition to the expected performance of personalized models, there are also works aiming at addressing the problems personalized models may meet when applied in reality. Ditto (Li et al., 2021c) adds the regularizer measuring the difference between personalized models and the global model into the objective functions to guarantee both the fairness and robustness of personalized models.

Apart from the usability of personalized models, the accessibility of personalized models is also a key consideration when it comes to real-world applications. Considering the situations where clients have heterogeneous environments like datasets, hardware, software, and the Internet, there are too many unpredictable situations in the real world blocking the access of personalized models. To solve these problems, some works (Wu et al., 2020; Li et al., 2021a) propose to allow clients to learn different personalized model structures. LotteryFL (Li et al., 2021a) proposes to let clients individually learn a lottery model, which is a subset of the global model. During the communication, these lottery models will be shared between servers and clients. Without the requirement of communicating a global model, this method can significantly reduce the communication cost in its training process. pFedHN (Shamsian et al., 2021) also makes clients learn a sub-model based on the global model.

Recently, there have also been many works exploring personalizing parts of models instead of the whole model to improve the performance of personalized models. LG-FedAvg (Liang et al., 2020) proposes to share the upper layers (model head) in the DNN and personalize the bottom layers (base model), which will not be averaged during the training. It utilizes personalized base models to output

different local features in different clients, on which the global model head will be collaboratively trained through the FedAvg. Conversely, FedRep (Collins et al., 2021) proposes to learn a global feature extractor and personalized classifiers. FedRoD (Chen & Chao, 2021) proposes a two-predictor framework in which clients train different model heads to switch between GFL and PFL.

Different from their work, Our framework considers a more challenging FL setting, i.e. every client may meet with OOD test data from other clients. Moreover, instead of improving the performance of the model itself, we consider more about how clients collaborate to handle the unpredictable test data.

### A.3 Incentive Mechanism

The purpose of FL collaboration among clients is the improvement of model performance on the test data. Therefore, it is important to know how much performance gain can be obtained after FL collaboration (Ghorbani & Zou, 2019; Liu et al., 2022; Sim et al., 2020). Furthermore, there should be a well-designed incentive mechanism (Ng et al., 2020; Yu et al., 2020a; Zeng et al., 2022) that motivates clients to join FL. Our modular store essentially provides a market economy to let clients autonomously choose and download the needed plug module. The higher the generalization performance of the plug-in module, the more favorable it is. Therefore, the incentive mechanism of the modular store is naturally connected with the practical benefits of the plug-in module.

### A.4 Federated Continual Learning

Continual learning (CL) (Kirkpatrick et al., 2017) is to learn different tasks sequentially. Some former tasks are inaccessible after training. Thus, when training subsequent tasks, the machine learning model may forget previous tasks. EWC (Kirkpatrick et al., 2017) finds the model parameters that are good for both previous and subsequent tasks using the Fisher Information Matrix. Progressive Neural Network approach (Rusu et al., 2016) is to increasingly construct the model during the training. Thus, the newly added parameters can learn the new tasks, while the old parameters can remember the old tasks. DEN (Yoon et al., 2018) dynamically decides the model capacity to learn a compact overlapping knowledge sharing among tasks.

Federated Continual Learning (FCL) (Yoon et al., 2021) is a new problem where, after FL training on some clients, there are some other clients that come and join the FL training. The trained model needs to avoid forgetting the previous dataset while performing well on the later dataset with data from newly arrived clients. We use a simple example to show that HPFL is naturally suitable to address the forgetting problem of FCL. Our plug-in can not only be seen as a personalized part of the model helping clients do inference on test data but also considered as a container preserving knowledge obtained from training. So it is natural to think we can store the knowledge in the previous dataset and access it whenever we are in need. In fact, it can be seen as an application of HPFL on the temporal scale. Most of the works in FL talk about many clients in a single period of time, i.e. Federated Learning in the spatial scale. FCL itself can be seen as a problem that happens at Federated Learning within the temporal scale: clients from different times collaboratively learn models that can generalize well on circumstances varied with time. We experimentally verified the potential of HPFL to address the forgetting problem of FCL in Section 5.4. Details about that experiment and more discussion are presented in Appendix D.5.

### A.5 Asynchronous FL

Asynchronous FL (Async-FL) (Xie et al., 2019) means to ease the constraint of the synchronous communication mechanism of classic federated optimization schemes (McMahan et al., 2017). In Async-FL, clients may download the global model from and return gradients to the server at different times. Thus, the server may receive a stale model update, causing unstable convergence. Such a staleness problem has long existed in the distributed machine learning area (Langford et al., 2009; Zheng et al., 2017). Stale updates are usually controlled by some staleness coefficients (Xie et al., 2019) or compensated by (Zheng et al., 2017) other newer gradients. Anarchic FL (Yang et al., 2022) can be seen as a more extreme version of Async-FL. In Anarchic FL, clients can decide to download and upload the models at any time, not controlled by the server at all. To this end, HPFL naturally allows this kind of working paradigm since once an average backbone, which can be obtained from pre-trained models or summoning several active clients to train, is accessible, any

aggregation operation is not in demand for HPFL, so the server doesn't rely on timely respond of client and won't be disturbed by stale model update. Once a plug-in is updated by the client, the plug-in can be utilized to do inference on appropriate test data without concern that the parameter of the model will change over time.

### A.6 TEST-ADAPTATION & DOMAIN ADAPTATION METHODS IN FL

There also emerge works that aim to adapt or generalize to new unseen clients with seen or unseen data distribution. FADA (Peng et al., 2019) utilize domain adaptation to tackle with seen target distribution. However, their method requires target domain data to train an adversarial model and thus cannot handle the situation where the target domain is unknown. FedDG (Liu et al., 2021) first proposed a novel setting where a federated model trained on multiple distributed source domains is required to generalize on unseen target domains. However, these methods all aim to train a unified global model for adaptation or generalization to new clients. As far as we know, HPFL is the first FL framework to directly exploit PMs to achieve this goal. FedTHE & FedTHE+ (Jiang & Lin, 2023) discuss test-time distribution shift, which is similar to our problem setting. However, we narrow down the category of distribution shift to apply to the GFL setting and perform much better in our proposed circumstance, while their method mainly aims at dealing with unknown distribution shift.

## B PROOF

### B.1 LOWER BOUND OF PM WITH GFL

**Theorem B.1.** *With Equation 4 and the PMs obtained from optimizing Equation 2 as:* $\Omega^{pfl}, \theta_1^{pfl}, ..., \theta_M^{pfl} = \arg\min_{\Omega,\theta_1,...,\theta_M} \mathcal{L}_P(\Omega, \theta_1, ..., \theta_M)$*, we have*

$$\mathcal{L}_{P-G}(\Omega, \theta_1, ..., \theta_M) \geq \mathcal{L}_P(\Omega^{pfl}, \theta_1^{pfl}, ..., \theta_M^{pfl}).$$

*Proof.*

$$\begin{aligned}
\mathcal{L}_{P-G}(\Omega, \theta_1, ..., \theta_M) &= \frac{1}{M}\sum_{i=1}^{M}\sum_{m=1}^{M} p_m \mathbb{E}_{\xi_m \sim \mathcal{D}_m} \ell(f(\theta_i, \xi_m), \xi_m) + \mathcal{R}(\Omega, \theta_1, ..., \theta_M) \\
&\geq \frac{1}{M}\sum_{i=1}^{M}\sum_{m=1}^{M} p_m \mathbb{E}_{\xi_m \sim \mathcal{D}_m} \ell(f(\theta_m, \xi_m), \xi_m) + \mathcal{R}(\Omega, \theta_1, ..., \theta_M) \\
&= \sum_{m=1}^{M} p_m \mathbb{E}_{\xi_m \sim \mathcal{D}_m} \ell(f(\theta_m, \xi_m), \xi_m) + \mathcal{R}(\Omega, \theta_1, ..., \theta_M) \\
&\geq \sum_{m=1}^{M} p_m \mathbb{E}_{\xi_m \sim \mathcal{D}_m} \ell(f(\theta_m^{pfl}, \xi_m), \xi_m) + \mathcal{R}(\Omega^{pfl}, \theta_1^{pfl}, ..., \theta_M^{pfl}) \\
&= \mathcal{L}_P(\Omega^{pfl}, \theta_1^{pfl}, ..., \theta_M^{pfl}),
\end{aligned}$$

which completes the proof. $\qquad\square$

### B.2 THE EQUIVALENCE BETWEEN SFL AND PFL

**Theorem B.2.** *With Equation 4,* $\Omega^{pfl}, \theta_1^{pfl}, ..., \theta_M^{pfl} = \arg\min_{\Omega,\theta_1,...,\theta_M} \mathcal{L}_P(\Omega, \theta_1, ..., \theta_M)$ *and the* $\mathcal{H}^*$ *that guarantees* $\theta_m^{pfl} = s(\Theta, \xi_m, \mathcal{H})$*, we have*

$$\mathcal{L}_{P-G}(\Omega, \theta_1, ..., \theta_M) \geq \mathcal{L}_P(\Theta) \geq \mathcal{L}_S(\Theta, \mathcal{H}^*).$$

*Proof.*

$$\mathcal{L}_S(\Theta, \mathcal{H}^*) = \sum_{m=1}^{M} p_m \mathbb{E}_{\xi_m \sim \mathcal{D}_m} \ell(\widehat{f}(\Theta, \xi_m, \mathcal{H}^*), \xi_m)$$

$$= \sum_{m=1}^{M} p_m \mathbb{E}_{\xi_m \sim \mathcal{D}_m} \ell(f(\theta_m^{pfl}, \xi_m), \xi_m)$$

$$= \mathcal{L}_P(\Omega^{pfl}, \theta_1^{pfl}, ..., \theta_M^{pfl}) - \mathcal{R}(\Omega^{pfl}, \theta_1^{pfl}, ..., \theta_M^{pfl})$$

$$\leq \mathcal{L}_P(\Omega^{pfl}, \theta_1^{pfl}, ..., \theta_M^{pfl}),$$

combining with Theorem 3.1, which completes the proof. □

## C   EXPERIMENT CONFIGURATION

### C.1   HARDWARE AND SOFTWARE CONFIGURATION

We conduct experiments using NVIDIA A100 40GB GPU, AMD EPYC 7742 64-Core Processor Units. The operating system is Ubuntu 20.04.1 LTS. The pytorch version is 1.12.1. The numpy version is 1.23.2. The cuda version is 12.0.

### C.2   CODE AND INSTRUCTIONS FOR REPRODUCIBILITY.

To ensure privacy, the codes and instructions will be uploaded as an anonymous link during the rebuttal phase.

### C.3   IMPLEMENT OF SIMPLIFIED METRICS AND PROOF

The original metric under the GFL-PM setting in classification tasks should be:

$$\text{Accuracy}\,(\Omega, \theta_1, \ldots, \theta_M) = \frac{1}{M} \sum_{i=1}^{M} \sum_{m=1}^{M} p_m \mathbb{E}_{\xi_m \sim \mathcal{D}_m} \mathcal{T}\,(f\,(\theta_i, \xi_m)\,, \xi_m) \tag{9}$$

where $\mathcal{T}(\cdot, \cdot)$ is the function judging whether the prediction of the model is the same with the real label, specifically

$$\mathcal{T}(\text{prediction}, \text{sample}) = \mathbb{1}(\text{predciton} = y_{sample}) \tag{10}$$

where $y_{sample}$ is the label of sample. $f\,(\theta_i, \xi_m)$ is the prediction of the model used in final inference on client i for the sample $\xi_m$, the model is parameterized with $\theta_i$. And our way of determining personalized model using when inferencing on client $i$ is to select from all the plug-ins, i.e. the PMs obtained from optimizing Equation 2 on local data: $\Omega^{pfl}, \theta_1^{pfl}, ..., \theta_M^{pfl} = \arg\min_{\Omega, \theta_1, ..., \theta_M} \mathcal{L}_P(\Omega, \theta_1, ..., \theta_M)$, so we have

$$\theta_i = \theta_{C_i(\xi_m, i)}^{pfl}, m \in 1, 2, ..., M \tag{11}$$

where $C_i(\xi_m, i)$ is the selection made for client i based on test data $\xi_m$ and client i. $C_i$ is the selection algorithm of the client i.

For traditional personalized methods, the clients will only use personalized models trained locally, i.e.

$$C_i(\xi_m, i) = i \tag{12}$$

substitute Equation 12 into Equation 11, we have

$$\theta_i = \theta_{C_i(\xi_m, i)}^{pfl} = \theta_i^{pfl} \tag{13}$$

then substitute Equation 13 into Equation 9, we have

$$
\begin{aligned}
\text{Accuracy}\left(\Omega, \theta_1, \ldots, \theta_M\right) &= \frac{1}{M} \sum_{i=1}^{M} \sum_{m=1}^{M} p_m \mathbb{E}_{\xi_m \sim \mathcal{D}_m} \mathcal{T}\left(f\left(\theta_i, \xi_m\right), \xi_m\right) \\
&= \frac{1}{M} \sum_{i=1}^{M} \sum_{m=1}^{M} p_m \mathbb{E}_{\xi_m \sim \mathcal{D}_m} \mathcal{T}\left(f\left(\theta_i^{pfl}, \xi_m\right), \xi_m\right) \\
&= \frac{1}{M} \sum_{i=1}^{M} \sum_{i=1}^{M} \sum_{m=1}^{M} p_m \frac{1}{n_m} \sum_{j=1}^{n_m} \mathcal{T}\left(f\left(\theta_i^{pfl}, \xi_j\right), \xi_j\right) \\
&= \frac{1}{M} \sum_{i=1}^{M} \sum_{m=1}^{M} \frac{n_m}{N} \frac{1}{n_m} \sum_{j=1}^{n_m} \mathcal{T}\left(f\left(\theta_i^{pfl}, \xi_j\right), \xi_j\right) \\
&= \frac{1}{MN} \sum_{i=1}^{M} \sum_{m=1}^{M} \sum_{j=1}^{n_m} \mathcal{T}\left(f\left(\theta_i^{pfl}, \xi_j\right), \xi_j\right) \\
&= \frac{1}{MN} \sum_{i=1}^{M} \sum_{j=1}^{N} \mathcal{T}\left(f\left(\theta_i^{pfl}, \xi_j\right), \xi_j\right) \\
&= \frac{1}{M} \sum_{i=1}^{M} \underbrace{\mathbb{E}_{\xi_D \sim \mathcal{D}}[\mathcal{T}\left(f\left(\theta_i^{pfl}, \xi_D\right), \xi_D\right)]}_{\text{accuracy of PM in client } i \text{ on global data}}
\end{aligned}
\tag{14}
$$

Equation 14 represents **the averaged accuracy of all personalized models on the global dataset**, so we can calculate the averaged accuracy of all personalized models on the global dataset as the metrics of simplified metrics instead of original complicated metrics 9; while for our proposed methods HPFL, because all clients have same selection method $C$

$$
C_i(\xi_m, i) = \arg \max_n g\left(\xi_n, \xi_m\right) = C(\xi_m)
\tag{15}
$$

the origin metric turns into

$$
\begin{aligned}
\text{Accuracy}\left(\Omega, \theta_1, \ldots, \theta_M\right) &= \frac{1}{M} \sum_{i=1}^{M} \sum_{m=1}^{M} p_m \mathbb{E}_{\xi_m \sim \mathcal{D}_m} \mathcal{T}\left(f\left(\theta_{argmax_n g(\xi_n, \xi_m)}, \xi_m\right), \xi_m\right) \\
&= \frac{1}{M} \sum_{i=1}^{M} \sum_{m=1}^{M} p_m \mathbb{E}_{\xi_m \sim \mathcal{D}_m} \mathcal{T}\left(f\left(\theta_{C(\xi_m)}^{pfl}, \xi_m\right), \xi_m\right) \\
&= \frac{1}{M} \sum_{i=1}^{M} \sum_{i=1}^{M} p_m \frac{1}{n_m} \sum_{j=1}^{n_m} \mathcal{T}\left(f\left(\theta_{C(\xi_m)}^{pfl}, \xi_j\right), \xi_j\right) \\
&= \frac{1}{M} \sum_{i=1}^{M} \sum_{m=1}^{M} \frac{n_m}{N} \frac{1}{n_m} \sum_{j=1}^{n_m} \mathcal{T}\left(f\left(\theta_{C(\xi_m)}^{pfl}, \xi_j\right), \xi_j\right) \\
&= \frac{1}{MN} \sum_{i=1}^{M} \sum_{m=1}^{M} \sum_{j=1}^{n_m} \mathcal{T}\left(f\left(\theta_{C(\xi_m)}^{pfl}, \xi_j\right), \xi_j\right) \\
&= \frac{1}{N} \sum_{m=1}^{M} \sum_{j=1}^{n_m} \mathcal{T}\left(f\left(\theta_{C(\xi_m)}^{pfl}, \xi_j\right), \xi_j\right) \\
&= \sum_{m=1}^{M} \frac{n_m}{N} \frac{1}{n_m} \sum_{j=1}^{n_m} \mathcal{T}\left(f\left(\theta_{C(\xi_m)}^{pfl}, \xi_j\right), \xi_j\right) \\
&= \sum_{m=1}^{M} p_m \frac{1}{n_m} \sum_{j=1}^{n_m} \mathcal{T}\left(f\left(\theta_{C(\xi_m)}^{pfl}, \xi_j\right), \xi_j\right)
\end{aligned}
$$

$$= \sum_{m=1}^{M} p_m \mathbb{E}_{\xi_m \sim \mathcal{D}_m} \mathcal{T}\left(f\left(\theta_{C(\xi_m)}^{pfl}, \xi_j\right), \xi_j\right)$$

$$= \sum_{i=1}^{M} p_i \underbrace{\mathbb{E}_{\xi_i \sim \mathcal{D}_i} \mathcal{T}\left(f\left(\theta_{C(\xi_i)}^{pfl}, \xi_j\right), \xi_j\right)}_{\text{accuracy of PM selected by client i on its own data}}, \quad (16)$$

Equation 16 represents the averaged accuracy of clients testing on their own personalized dataset with models equipped with their selected plug-ins based on their own data, weighted with number of samples in data on clients. With these simplification of metrics, we can more efficiently test GFL-PM performance of both the traditional personalized methods (FedPer, FedRoD, FedRep) and HPFL.

### C.4 HYPER-PARAMETERS

We use SGD without momentum as the optimizer for all experiments, with a batch size of 128 and weight decay of 0.0001. The learning rate is set as 0.1 for both the training of the global model and the fine-tuning on local datasets. The main results shown in Tabel 2 are conducted with 1-layer plug-ins (i.e. only classifier).

Special hyperparameters of some baseline methods are :

**FedRep:** Local personalize epoch is set as 1.

**PerFedMask:** The partition percent of validation is 0.1, personalized fine-tuning epoch $E_p$ after calculating mask is set as 5 as the official implementation did.

**FedTHE:** We follow the official implementation: the smoothing factor of test history descriptor maintained by the Exponential Moving Average (EMA) $\alpha$ equals 0.1; the smoothing factor interpolating the test feature and the test history descriptor $\beta$ equals 0.3.

**FedSAM:** We follow the official implementation: the parameters for SAM minimizers $\rho = 0.1$ , $\eta = 0$.

### C.5 EXTRA EXPLANATION ON EXPERIMENT

Due to the limited space of the main text, we show a more detailed explanation of the experiment in this section.

For the construction of the personalized test dataset, to make the training data and test data of a client have the same distribution following the settings of most PFL methods (Collins et al., 2021), we count the number of samples $S_{train}(c, m)$ in each class c of training data of client $m$ and split test data of that clients in that distribution (which means client m have $\frac{S_{train}(c,m)}{\sum_{m=1}^{N}\sum_{c=1}^{C} S_{train}(c,m)} \times \sum_{m=1}^{M}\sum_{c=1}^{C} S_{test}(c,m)$ test samples in class $c$, here $C$ denotes the number of classes in overall dataset, $M$ denotes the number of clients in the FL system), Figure 5 and Figure 6 shows that the data partition of training data and test data are almost identical as expected in PFL.

To report the best result of all baseline methods, we report the accuracy of their best inference global model on global data during the whole training process. For our method, we also use the best inference model as the backbone of HPFL for fair comparison.

## D EXTRA EXPERIMENT RESULTS

Due to the limited space of the main text, we show more experiment results in this section.

### D.1 MORE RESULTS ABOUT DIFFERENT NUMBER OF LAYERS OF PLUG-IN

In this part, we are trying to explain why the selection method degrades when the number of plug-in layers increases in an intuitive way. We propose two possible ways leading to the degradation: (1) selection methods can't handle the large dimension of features that vastly increases when the number of plug-in layers increases, as shown in Table 6. (2) A larger number of plug-in layers means we are

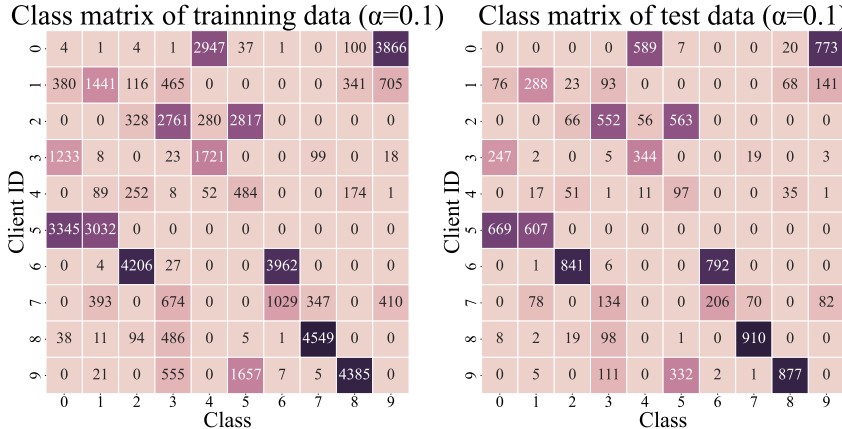

Figure 5: Data partitioning on CIFAR-10 ($\alpha$=0.1)

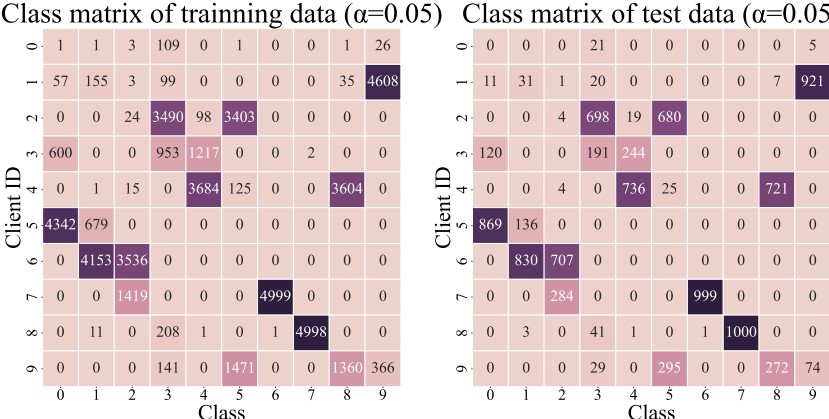

Figure 6: Data partitioning on CIFAR-10 ($\alpha$=0.05)

using features extracted with more shallow layers, and these features tend to be more local, which may not be so helpful for the selection method to assess the similarity of the distributions they are sampled from. For further study, we may conduct experiments to testify these two conjectures. Once the conjectures are testified, we will try to find ways to solve these two problems. However, despite the difficulty of choosing, large plug-ins also multiply the computation time and resources needed in training them, the network bandwidth required to transmit them, and so on. As a result, large plug-ins are generally not good options in HPFL from our perspective.

We also explore selection with different numbers of layers of plug-ins in different settings, like $\alpha$=0.05, M=10 on CIFAR-10. Figure 4 and Figure 7 show that with the number of plug-in layers going up, the selection becomes more difficult and unstable as we claimed before.

Table 6: # feature dimensions versus # plug-ins layers on CIFAR-10.

| # plug-ins layers | 1 | 3 | 4 | 5 | 6 |
|---|---|---|---|---|---|
| # feature dimensions | 512 | 512×4×4 (8,192) | 256×8×8 (16,384) | 128×16×16 (32,768) | 64×32×32 (131,072) |

## D.2 MORE RESULTS ABOUT SELECTION METHOD (SVCCA, CKA, OOD)

**SVCCA and CKA.** SVCCA (Raghu et al., 2017) exploits singular value decomposition and canonical correlation analysis to compare the features learned by different DNNs. CKA (Kornblith et al., 2019)

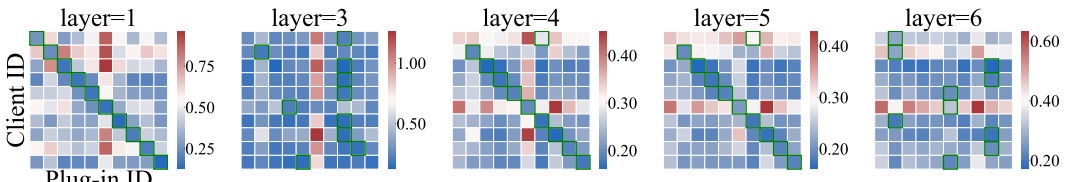

Figure 7: Selection score maps with different number of plug-in layers on CIFAR-10 ($\alpha = 0.05$)

utilizes the normalized HSIC (Gretton et al., 2005) to measure the similarity. CKA is invariant to the invertible linear transformation. Thus, it can measure meaningful similarities between representations of high dimension (Kornblith et al., 2019).

**OOD detection based methods.** Out-of-distribution (OOD) detection aims to find out whether the test data is OOD or not. Current OOD detection methods includes the norm of gradients (Huang et al., 2021), distance-based methods (Sun et al., 2022), reconstruction based methods (Zhou, 2022), classifier based methods (Katz-Samuels et al., 2022; Du et al., 2022). Intuitively, if the test data is OOD to one plug-in, we can discard this plug-in. Thus, based on this insight, we train an OOD classifier $\tau_m$ for each plug-in $\rho_m$. Each OOD classifier will output an OOD confidence $\tau_m(h_{test})$. Lower $\tau_m(h_{test})$, less possible that $h_{test}$ is OOD to the plug-in $\rho_m$. Then, the select problem turns from Equation 6 into:

$$S_{OOD}(h_{test}, \tau_1, ..., \tau_M) = \arg\min_{i \in \mathcal{M}} \tau_i(h_{test}). \tag{17}$$

The $\tau_m$ is trained during the process of optimizing $\rho_m$. On each client $m$, we generate random noise $\epsilon \sim \mathcal{N}(0, \mu_\epsilon)$ as the OOD data, the $h_m$ is seen as the in-distribution (ID) data. OOD data has label 1, and ID data has label 0. We use a linear classifier as the OOD classifier $\tau_m$ whose input dimension is same with $h_m$ and the output dimension is 1. For OOD data, we hope the $\tau_m$ outputs 1. We use the cross-entropy loss to train the OOD classifier. Different from the distance based methods, the OOD detection based method does not need to communication the processed hidden features $\hat{h}_m$, which significantly increases the privacy security.

We show the results of our previous attempts using SVCCA, CKA, and OOD as the selection methods in this section. The overall training and inference process is the same as mentioned in Section 4. When implementing SVCCA in $\mathcal{M}$=100, we encountered the problem that the number of samples in a client was not enough for SVCCA, which required at least $N_{component}$ components. To deal with it, we change the number of components used to calculate SVCCA similarity between the noised features of training data and the features of test data to min(min(num_sample), $N_{component}$). From Table 7 and Figure 8, it is easy to conclude that SVCCA, CKA (with 4 kinds of kernel), OOD all failed to select correct plug-ins. Take HPFL based on OOD detection as an example, OOD classifiers failed to give good predictions on whether test data is OOD or not for plug-ins because in federated learning, it is difficult to get OOD data as negative samples when training locally, which will cause OOD classifiers only see in-distribution(ID) data. We tried to solve this problem by generating random images as OOD data to train the OOD classifiers. It is easy to see this method did not work as shown in Figure 8. Therefore, it is difficult to tell whether there is an appropriate way to train OOD classifiers to determine whether OOD detection can be utilized in HPFL. There we propose a possible way to train OOD classifiers: after clients upload the noised features of training data to the server, it is possible to utilize these features to train a good OOD classifier and select plug-ins using these classifiers. We take this as a future direction in exploring more selection methods that can be used in HPFL.

### D.3    MORE RESULTS ABOUT SELECTION ACCURACY

In this part, we take a closer look at the selection accuracy using MMD by visualizing all the selection situation on CIFAR-10, FMNIST, CIFAR-100, with three settings {$\alpha$=0.1, $\mathcal{M}$=10}, {$\alpha$=0.05, $\mathcal{M}$=10}, {$\alpha$=0.1, $\mathcal{M}$=100} as arrange from left to right in Figure 9, 10, 11. The colors of the blocks denote the MMD scores, where red represents a relatively high score, and blue represents a relatively low score. The green box on the block(i,j) implies that client i is choosing plug-in j with minimal MMD score

Table 7: Experiment results of HPFL using SVCCA, CKA and OOD. Noisy coefficient $\kappa=1$, FedAvg is fine-tuned with the whole model instead of only part of model as in HPFL.

| Clients | 10 | | | | | | 100 | | |
|---|---|---|---|---|---|---|---|---|---|
| Non-IID | Dir(0.1) | | | Dir(0.05) | | | Dir(0.1) | | |
| Test Set | GFL | | PFL | GFL | | PFL | GFL | | PFL |
| Method/Model | GM | PM | PM | GM | PM | PM | GM | PM | PM |
| **CIFAR-10** | | | | | | | | | |
| HPFL(SVCCA) $E_p = 1$ | 81.5 | 62.8 | 95.4 | 62.4 | 32.7 | 96.0 | 73.6 | 61.0 | 95.0 |
| HPFL(SVCCA) $E_p = 10$ | 81.5 | 62.5 | 95.8 | 62.4 | 34.7 | 96.3 | 73.6 | 47.0 | 95.7 |
| HPFL(Linear-CKA) $E_p = 1$ | 81.5 | 56.1 | 95.4 | 62.4 | 55.2 | 96.0 | 73.6 | 70.5 | 95.0 |
| HPFL(Linear-CKA) $E_p = 10$ | 81.5 | 55.1 | 95.8 | 62.4 | 44.6 | 96.3 | 73.6 | 60.3 | 95.7 |
| HPFL(RBF-CKA) $E_p = 1$ | 81.5 | 61.0 | 95.4 | 62.4 | 55.2 | 96.0 | 73.6 | 70.7 | 95.0 |
| HPFL(RBF-CKA) $E_p = 10$ | 81.5 | 55.9 | 95.8 | 62.4 | 44.6 | 96.3 | 73.6 | 59.9 | 95.7 |
| HPFL($Linear - CKA_{debias}$) $E_p = 1$ | 81.5 | 63.9 | 95.4 | 62.4 | 47.2 | 96.0 | 73.6 | 66.3 | 95.0 |
| HPFL($Linear - CKA_{debias}$) $E_p = 10$ | 81.5 | 59.0 | 95.8 | 62.4 | 37.7 | 96.3 | 73.6 | 53.0 | 95.7 |
| HPFL($RBF - CKA_{debias}$) $E_p = 1$ | 81.5 | 61.0 | 95.4 | 62.4 | 35.0 | 96.0 | 73.6 | 68.4 | 95.0 |
| HPFL($RBF - CKA_{debias}$) $E_p = 10$ | 81.5 | 56.7 | 95.8 | 62.4 | 40.2 | 96.3 | 73.6 | 55.7 | 95.7 |
| HPFL(OOD) $E_p = 1$ | 81.5 | 66.3 | 95.4 | 62.4 | 54.4 | 96.0 | 73.6 | 64.0 | 95.0 |
| HPFL(OOD) $E_p = 10$ | 81.5 | 16.0 | 95.8 | 62.4 | 3.9 | 96.3 | 73.6 | 27.9 | 95.7 |
| **FMNIST** | | | | | | | | | |
| HPFL(SVCCA) $E_p = 1$ | 86.0 | 61.8 | 98.3 | 76.1 | 41.7 | 99.0 | 90.2 | 90.0 | 97.9 |
| HPFL(SVCCA) $E_p = 10$ | 86.0 | 49.7 | 98.4 | 76.1 | 49.3 | 99.2 | 90.2 | 87.7 | 98.8 |
| HPFL(Linear-CKA) $E_p = 1$ | 86.0 | 67.6 | 98.3 | 76.1 | 73.2 | 99.0 | 90.2 | 89.1 | 97.9 |
| HPFL(Linear-CKA) $E_p = 10$ | 86.0 | 62.0 | 98.4 | 76.1 | 65.5 | 99.2 | 90.2 | 88.6 | 98.8 |
| HPFL(RBF-CKA) $E_p = 1$ | 86.0 | 67.6 | 98.3 | 76.1 | 73.0 | 99.0 | 90.2 | 89.4 | 97.9 |
| HPFL(RBF-CKA) $E_p = 10$ | 86.0 | 63.3 | 98.4 | 76.1 | 65.5 | 99.2 | 90.2 | 88.7 | 98.8 |
| HPFL($Linear - CKA_{debias}$) $E_p = 1$ | 86.0 | 66.6 | 98.3 | 76.1 | 44.7 | 99.0 | 90.2 | 89.9 | 97.9 |
| HPFL($Linear - CKA_{debias}$) $E_p = 10$ | 86.0 | 51.8 | 98.4 | 76.1 | 37.7 | 99.2 | 90.2 | 88.3 | 98.8 |
| HPFL($RBF - CKA_{debias}$) $E_p = 1$ | 86.0 | 61.2 | 98.3 | 76.1 | 50.7 | 99.0 | 90.2 | 89.9 | 97.9 |
| HPFL($RBF - CKA_{debias}$) $E_p = 10$ | 86.0 | 51.8 | 98.4 | 76.1 | 27.9 | 99.2 | 90.2 | 87.6 | 98.8 |
| HPFL(OOD) $E_p = 1$ | 86.0 | 83.9 | 98.3 | 76.1 | 73.6 | 99.0 | 90.2 | 87.2 | 97.9 |
| HPFL(OOD) $E_p = 10$ | 86.0 | 42.7 | 98.4 | 76.1 | 37.5 | 99.2 | 90.2 | 88.4 | 98.8 |
| **CIFAR-100** | | | | | | | | | |
| HPFL(SVCCA) $E_p = 1$ | 68.6 | 68.2 | 83.3 | 65.3 | 68.2 | 87.4 | 59.7 | 51.8 | 81.2 |
| HPFL(SVCCA) $E_p = 10$ | 68.6 | 55.2 | 85.7 | 65.3 | 55.2 | 88.8 | 59.7 | 39.9 | 84.1 |
| HPFL(Linear-CKA) $E_p = 1$ | 68.6 | 63.4 | 83.3 | 65.3 | 63.4 | 87.4 | 59.7 | 51.0 | 81.2 |
| HPFL(Linear-CKA) $E_p = 10$ | 68.6 | 55.0 | 85.7 | 65.3 | 55.0 | 88.8 | 59.7 | 40.0 | 84.1 |
| HPFL(RBF-CKA) $E_p = 1$ | 68.6 | 64.1 | 83.3 | 65.3 | 64.1 | 87.4 | 59.7 | 51.4 | 81.2 |
| HPFL(RBF-CKA) $E_p = 10$ | 68.6 | 50.9 | 85.7 | 65.3 | 50.9 | 88.8 | 59.7 | 38.8 | 84.1 |
| HPFL($Linear - CKA_{debias}$) $E_p = 1$ | 68.6 | 62.8 | 83.3 | 65.3 | 62.8 | 87.4 | 59.7 | 50.7 | 81.2 |
| HPFL($Linear - CKA_{debias}$) $E_p = 10$ | 68.6 | 52.6 | 85.7 | 65.3 | 52.6 | 88.8 | 59.7 | 38.8 | 84.1 |
| HPFL($RBF - CKA_{debias}$) $E_p = 1$ | 68.6 | 62.9 | 83.3 | 65.3 | 62.9 | 87.4 | 59.7 | 50.8 | 81.2 |
| HPFL($RBF - CKA_{debias}$) $E_p = 10$ | 68.6 | 55.9 | 85.7 | 65.3 | 55.9 | 88.8 | 59.7 | 38.4 | 84.1 |
| HPFL(OOD) $E_p = 1$ | 68.6 | 63.5 | 83.3 | 65.3 | 63.5 | 87.4 | 59.7 | 49.5 | 81.2 |
| HPFL(OOD) $E_p = 10$ | 68.6 | 66.7 | 85.7 | 65.3 | 66.7 | 88.8 | 59.7 | 46.5 | 84.1 |

for local test data. MMD always helps clients select plug-in trained on the client itself (the green boxes denoting final choice of plug-ins all lie on the diagonal of score heatmap) when $\alpha=0.1$, $\mathcal{M}=10$ and $\alpha=0.05$, $\mathcal{M}=10$ in CIFAR-10 and FMNIST as shown in Figure 9, Figure 10. When it comes to a FL system with more clients, like $\mathcal{M}=100$, even though there is some clients choosing inappropriate plug-ins, most of them can still choose plug-ins trained on their own data. However, When conducted on a more heterogeneous dataset CIFAR-100, judging which plug-ins to choose becomes a more difficult task, which can be easily observed in Figure 11, as the increasing number of green boxes not located on the diagonal is indicating worse plug-ins selection. Additionally, the score map of CIFAR-100 is overall much whiter than the score map of CIFAR-10 and FMNIST, denoting that the scores of different plug-ins are close to each other and is more challenging to choose from when MMD encountered with CIFAR-100. Thus, how to improve selection methods used in HPFL is a crucial problem when met with more heterogeneous data together with fewer samples on each client, and will be important for future work.

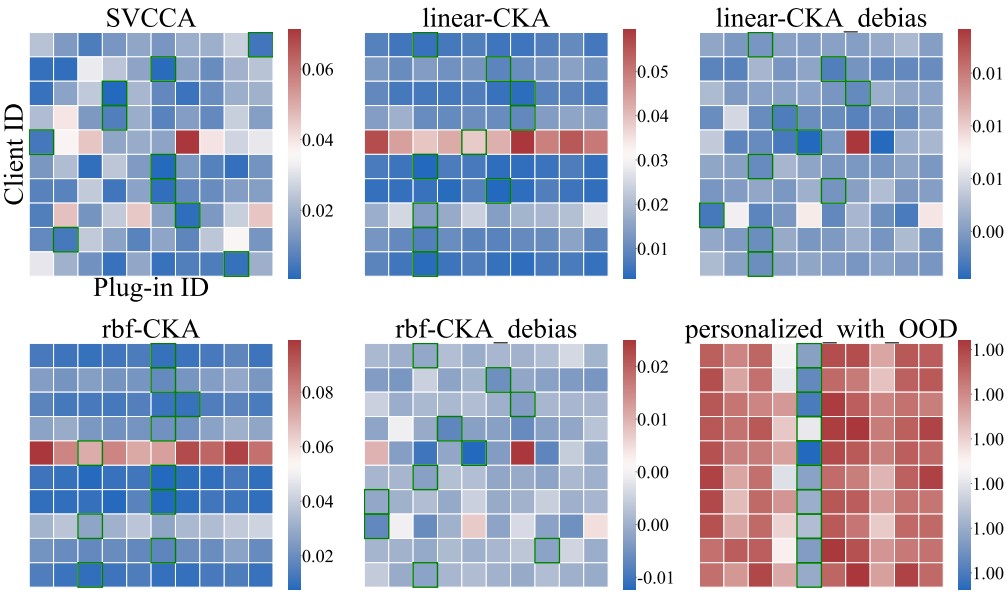

Figure 8: Selection score maps of SVCCA, CKA, OOD on CIFAR-10 ($\alpha = 0.1$)

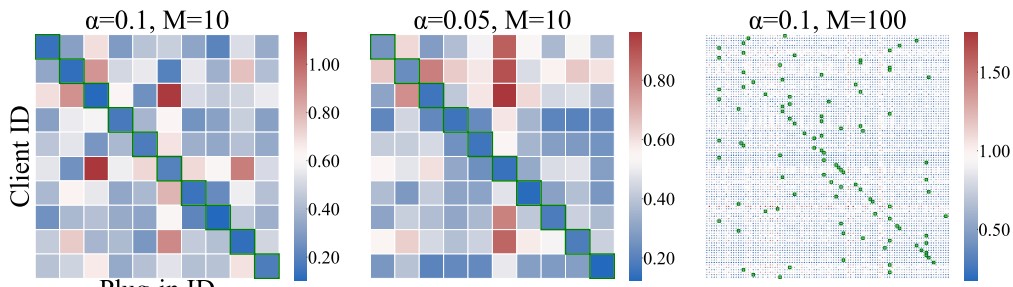

Figure 9: Selection score maps on CIFAR-10

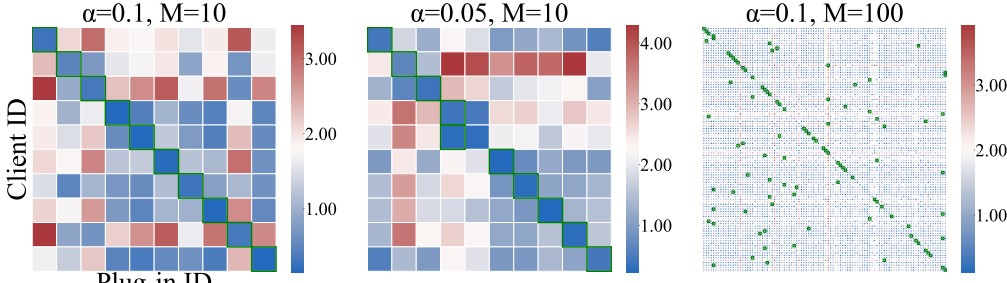

Figure 10: Selection score maps on FMNIST

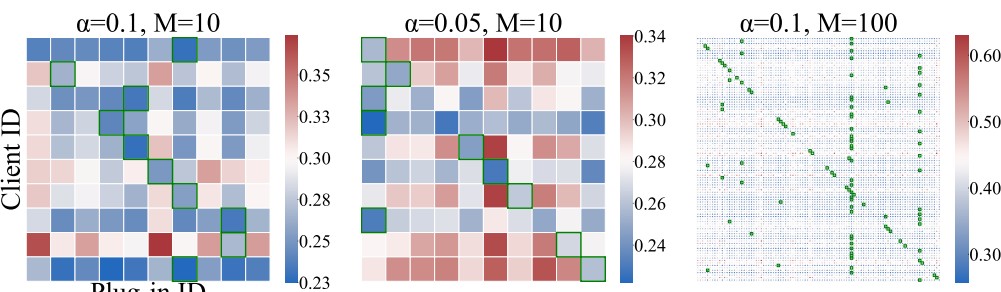

Figure 11: Selection score maps on CIFAR-100

### D.4 MORE RESULTS ABOUT NOISE

When using the HPFL methods based on MMD, it is required that the distribution of local features, or local features dealt with noise is transmitted together with the plug-in trained in the client, so that the other clients are able to select appropriate plug-ins based on these information. However, transmitting raw features is faced with the risk of data leakage when met with inversion attacks. In order to better protect privacy safety, we tried to add Gaussian noise generated with the distribution of local features to the origin features, surprisingly found that adding noise according to the distribution of the features not damage the performance, according to Table 8. Further study may transmit Gaussian noise generated with the distribution of the local features instead of the noised features. In fact, when $\kappa$ reaches a high value like 1000 in Figure 3, the noised features can be approximately considered to degenerate into the Gaussian noise. From Table 8, Figure 3 and Figure 12, we can observe increasing $\kappa$ to a large value doesn't hurt much performance of HPFL. Therefore, we will explore using pure Gaussian noise generated with the distribution of the local features to replace the noised features to better protect privacy in the future.

Table 8: Accuracy of Different noise coefficient $\kappa$ on CIFAR-10.

| Noise coefficient $\kappa$ | 0 | | 1 | | 10 | | 100 | | 1000 | |
|---|---|---|---|---|---|---|---|---|---|---|
| Fine-tune epoch $E_{tune}$ | 1 | 10 | 1 | 10 | 1 | 10 | 1 | 10 | 1 | 10 |
| $\alpha = 0.1, \mathcal{M} = 10$ | | | | | | | | | | |
| Accuracy | 95.4 | 95.7 | 95.4 | 95.7 | 95.4 | 95.7 | 95.4 | 95.7 | 95.4 | 95.7 |
| $\alpha = 0.05, \mathcal{M} = 10$ | | | | | | | | | | |
| Accuracy | 96.0 | 96.3 | 96.0 | 96.3 | 73.7 | 74.0 | 71.1 | 70.5 | 71.1 | 70.5 |
| $\alpha = 0.1, \mathcal{M} = 100$ | | | | | | | | | | |
| Accuracy | 95.4 | 92.0 | 95.4 | 95.7 | 95.4 | 95.7 | 95.4 | 95.7 | 95.4 | 95.7 |

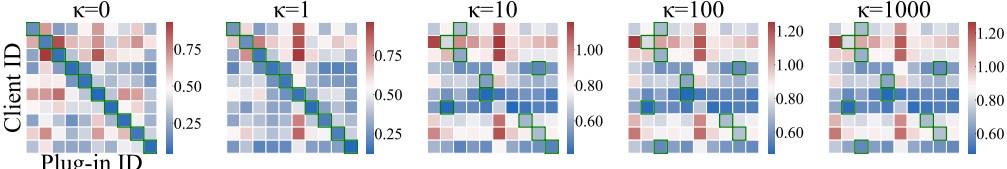

Figure 12: Selection score maps with different noise coefficient on CIFAR-10 ($\alpha$=0.05, $\mathcal{M}$=10)

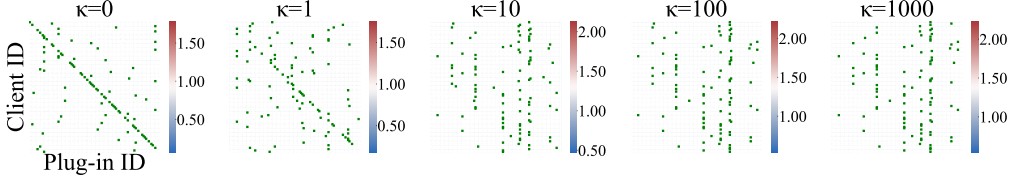

Figure 13: Selection score maps with different noise coefficient on CIFAR-10 ($\alpha$=0.1, $\mathcal{M}$=100)

### D.5 EXPERIMENT ABOUT FEDERATED CONTINUAL LEARNING

With the increasing real applications of Federated Learning, Federated Continual Learning (FCL) has attracted the attention of researchers. In this part, we conduct an experiment to display the potential of HPFL to solve catastrophic forgetting met in FCL. We first displayed the catastrophic forgetting issue in naive FCL. Then we utilized HPFL to solve this problem. Suppose we had 10 clients in the FL system. We first trained 500 epochs on client 0-4 with FedAvg, and then we trained another 500 epochs on client 5-9. We trained the backbone of HPFL and the global model of naive FCL in Nvidia V100 GPU and the rest of the experiment on Nvidia A100. For naive FCL, We had to adjust the learning rate to 0.05 when training on client 5-9 during 500-1000 epoch in case of training divergence. For FCL under HPFL, we froze the backbone of the model after training 500 epochs

on client 0-4 and training 5 plug-ins on client 0-4 for 1 epoch, respectively. Then we kept training on client 5-9 with the invariant backbone, after another 500 epochs, we trained 5 plug-ins on client 5-9, respectively. From Table 9, we observe that the accuracy of naive FCL significantly drops from 78.6 to 52.8, showing that training on clients 5-9 during 500-1000 rounds makes the global model severely forget the knowledge about clients 0-4. We show a promising way of using HPFL to mitigate this problem. When met with a new task, HPFL allows clients to quickly adapt to their local data by fine-tuning only a few epochs and uploading the plug-in to the server, like what happened at the 500 round in our experiment. After training in some new tasks, it is about time to conduct inference on all clients, we train plug-ins on new tasks, as we do on clients 5-9 in our experiment, and select plug-ins for every client. In that case, we are able to select and download the plug-ins better suited for test data with similar distribution, instead of having no choice but to use a global model having forgotten the knowledge of previous tasks. As is shown in Table 9, our experiment shows HPFL can significantly outperform naive FCL in GFL and mitigate the catastrophic forgetting issue in FCL.

Table 9: catastrophic forgetting issue in Naive FCL.

| Algorithm | Naive FCL (500R) | Naive FCL (1000R) |
|---|---|---|
| Test data | data from Client 0-4 | |
| Method/Model | GM | |
| Accuracy | 78.6 | 52.8 (↓ 25.8) |

### D.6 MORE RESULTS ABOUT BACKBONE TRAINING METHODS

As long as the used GFL methods are able to train a strong general feature extractor, HPFL is able to utilize the feature extractor to train the personalized plug-ins and extract features. We conduct experiments using FedRoD to testify HPFL's compatibility with other GFL methods. The results are given in Table 10. Number of clients equal $\mathcal{M} = 10$, local fine-tuning epoch $E_p = 10$, local datasets are partitioned in $Dir(0.1)$. Other settings remain the same as the main experiments in Table 2.

Table 10: Ablation study of backbone training methods.

| Clients | | 10 (sample 50% each round) | |
|---|---|---|---|
| Non-IID | | Dir(0.1) | |
| Test Setting | | GFL-PM | |
| Method | | HPFL(FedAvg) | HPFL(FedRoD) |
| CIFAR-10 | | | |
| | GFL-GM | 81.5 | 85.3 (↑ 3.8) |
| CIFAR-10 | GFL-PM | 95.7 | 96.0 (↑ 0.3) |
| | PFL-PM | 95.7 | 96.0 (↑ 0.3) |
| FMNIST | | | |
| | GFL-GM | 86.0 | 87.9 (↑ 1.9) |
| FMNIST | GFL-PM | 98.4 | 98.4 (↑ 0) |
| | PFL-PM | 98.4 | 98.4 (↑ 0) |
| CIFAR-100 | | | |
| | GFL-GM | 68.6 | 69.9 (↑ 1.3) |
| CIFAR-100 | GFL-PM | 72.2 | 68.5 (↓ 3.7) |
| | PFL-PM | 85.7 | 85.5 (↓ 0.2) |
| Tiny-ImageNet-200 | | | |
| | GFL-GM | 56.5 | 57.4 (↑ 0.9) |
| Tiny-ImageNet-200 | GFL-PM | 50.9 | 56.0 (↑ 5.1) |
| | PFL-PM | 73.7 | 74.7 (↑ 1.0) |

From the overall performance of HPFL(FedRoD), we can see that HPFL using FedRoD as its backbone training method is comparable to that using FedAvg, which confirms HPFL is compatible with the GFL methods other than FedAvg. We also observe an interesting fact that even if FedRoD shows excellent performance in GFL-GM (surpasses FedAvg in many datasets and settings), fine-tuning the backbone trained with it is not advantageous as shown in PFL-PM (only comparable with fine-tuning on the backbone trained with FedAvg). From this phenomenon, we presume the advantage of FedRoD in GFL-GM should mainly be attributed to its global head trained with a class-balanced loss instead of its backbone.

# E  DISCUSSION ON PRIVACY PROBLEM

As HPFL requires local clients to share auxiliary information on local data and plug-ins to help inference, it may raise concern about data privacy of HPFL. We attempt to analyze the risk of privacy leakage in HPFL respectively from sharing auxiliary information and plug-ins.

## E.1  PRIVACY RISKS OF SHARING PLUG-INS

In HPFL, we ask local clients to upload part of their personalized models to the server, which means every personalized model is possibly accessible to all clients. This potential sharing with other clients will raise concerns about the risk of privacy leakage. However, in classic Federated Learning algorithms like FedAvg, there also exists similar behavior of sharing global model, and it is difficult to recover training samples from the final model shared over the whole FL system. Instead, research shows that it is possible to recover training data of clients from gradients transmitted to the server (Geiping et al., 2020), which will not happen in HPFL except for the training period of the backbone model, which is able to be solved with regular privacy protect techniques like differential privacy (DP) which is widely used to protect potential privacy risks of GFL algorithms, and not a special problem of HPFL. Even extreme concern on potential privacy risk of storing plug-ins in the server can be solved by only requesting plug-ins after selection as described in Appendix F.2, clients providing the plug-ins can ask the server to delete the plug-ins after sending the plug-ins to the clients in need.

## E.2  PRIVACY RISKS OF SHARING AUXILIARY INFORMATION

Since HPFL asks clients to share auxiliary information with the server, once data breach happens in the communication period between clients and the server or the information is not properly kept in the server, The leaked information may lead to attacks, such as feature inversion attacks. Here we resorted image reconstruction by feature inversion method in (Zhao et al., 2020) to check whether the raw image can be reconstructed by inverting the representation through the pretrained global backbone model parameters, exploring whether data privacy will be threatened if both auxiliary information and backbone model is leaked. Experiments are conducted on CIFAR-10 with a ResNet-18 trained on CIFAR-10 under the Federated Learning setting used in our main experiments. As we can see in Figure 14, if we use raw features as the auxiliary information, real pictures can be easily recovered by feature inversion methods. To handle this risk of privacy leakage, here we propose three ways to prevent the problem: (1) add noise to the features; (2) use the averaged feature to select the plug-ins; (3) use model-based selection methods like OOD.

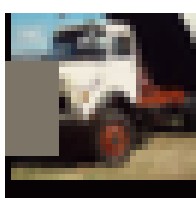 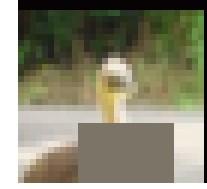 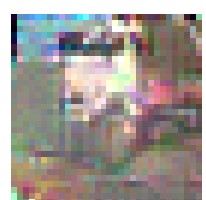 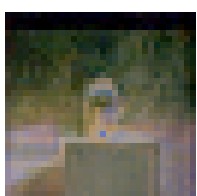

Raw image A        Raw image B        Reconstructed image A  Reconstructed image B

Figure 14: Image reconstructed from raw features

**Adding noise to transmitted information** is often practiced in the Federated Learning called Differential Privacy(DP), which is utilized to protect gradient against Differential attacks. Inspired

by DP, we attempt to add noise to the transmitted auxiliary information, and below we use the same recovery method to recover the original image from the noised features. We show some recovery results with the noised features in Figure 15, as noise coefficient $\kappa$ increases, the reconstructed image is less similar to the raw images, especially when $\kappa = 1$ as in our main experiment, actually the reconstructed images are hard to tell any information about the raw images. If there is still concern in this situation, the clients can increase the noise coefficient $\kappa$ to a higher level with the risk of lowering performance.

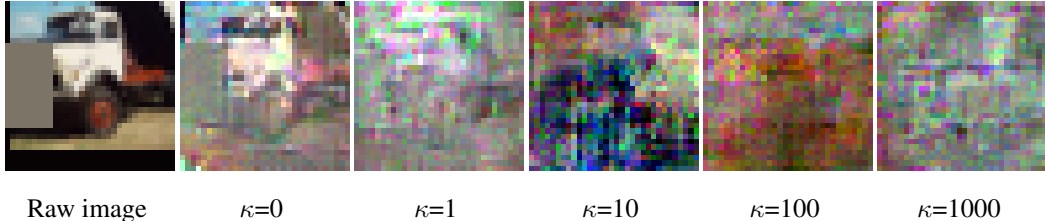

Raw image          $\kappa=0$          $\kappa=1$          $\kappa=10$          $\kappa=100$          $\kappa=1000$

Figure 15: Image reconstructed from the noised features, $\kappa$ is noise coefficient denoting the scale of noise added on raw features, the bigger $\kappa$ is, the larger noise is added on the features transmitted to select plug-ins.

**Using the averaged feature to select the plug-ins** is a practical way of protecting privacy as practiced in (Luo et al., 2021), inspired by their work, we attempted to select plug-ins with the average of all features on local clients. However, we assumed that simply averaging all features leads to the lack of information to select plug-ins properly, thus degrades the performance of HPFL. Therefore, we tried to divide the features into groups and take the average in every group. With enough samples in every group, we can prevent privacy leakage as presented in Figure 16 and maintain a good performance as shown in Table 11.

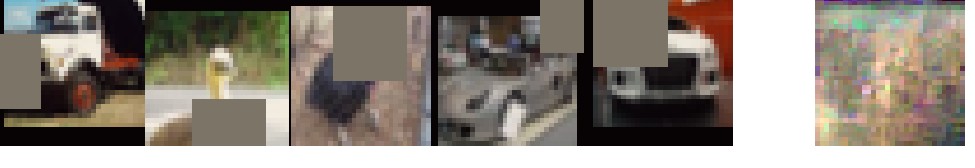

Raw images                                        Reconstructed image

Figure 16: Image reconstructed from the averaged feature, every group is composed by raw features of 10 raw images

Table 11: Accuracy of Different average group on CIFAR-10.

| # of raw features in every group | 3 | | 10 | |
|---|---|---|---|---|
| $E_p$ | 1 | 10 | 1 | 10 |
| $\alpha = 0.1, \mathcal{M} = 10$ | | | | |
| Accuracy | 81.5 | 80.1 | 76.8 | 79.1 |
| $\alpha = 0.05, \mathcal{M} = 10$ | | | | |
| Accuracy | 96.0 | 96.1 | 87.6 | 86.1 |
| $\alpha = 0.1, \mathcal{M} = 100$ | | | | |
| Accuracy | 81.4 | 83.8 | 76.8 | 75.3 |

**Utilizing model-based selection methods** like OOD to select the plug-ins, due to these methods avoid sharing direct information about raw data or features, they are exposed to less risk of data leakage. It is more difficult for the attacker to attack the clients with the model parameters than

with the data information due to less information contained in it, which can be proved by the data processing inequality (McMahan et al., 2017).

# F    REAL-WORLD APPLICATION

## F.1    REAL-WORLD GM-PFL

To better illustrate the GFL-PM setting we propose and demonstrate its importance, we give some examples exhibiting the significance of our proposed set-up below:

**Case 1:** Some clients may have insufficient computing resources or local training data to fine-tune a deep learning model in a cross-device setting. In these situations, training distribution can be regarded as an empty set $\emptyset$. In this way, the client cannot get a personalized model by locally fine-tuning the global model. In traditional GFL and PFL setting, the client has no choice but to adopt the global model and endure the lack of personalization. This problem is caused by the mismatch of training data distribution and test data distribution, as assumed in our proposed set-up, and is solvable with our proposed method HPFL by exploiting personalized plug-ins from other clients.

**Case 2:** A car with a personalized automated driving system (ADS) has driven out of the previous city it used to be. It requires to personalize on geometric data from the present city it is now in for improving the performance of the ADS in this new city. Classic GFL and PFL in this situation leave the ADS no option but to collect the geometric data and personalize on it after the collection completes, and accept the temporary performance loss using the previous personalized model before finishing the new personalization, since the distribution of test data has greatly changed. It's another example where the discrepancy between training data (geometric data from the previous city) and test data (geometric data from the present city) threatens the availability of FL systems. While with our proposed method designed to solve the problem, the ADS can attempt to access the plug-ins from car owners living in the present city.

**Case 3:** Imagine a person is traveling from a high latitude area to an equatorial region, and the recommender system on their phone is supported by federated learning. If the recommender system uses the personalized model trained when in the high latitude area, it will continue to prompt thick down jackets for the person, which is clearly an unexpected and unreasonable recommendation. With our method, one can get the same recommendation as the local people with plug-ins on their phones without time to fine-tune the model again.

## F.2    SCALABILITY ISSUES OF HPFL AND THEIR SOLUTIONS

### F.2.1    POTENTIAL SCALABILITY PROBLEM OF SHARING PLUG-INS

**For plug-ins:** In fact, HPFL can be applied to both cross-device and cross-silo setting, with a slight modification in cross-device setting where the number of clients is overwhelmingly large. We introduce the methods to enhance the scalability problem of HPFL as follows: To handle the massive plug-ins needed to be stored in the server, the server can cluster the plug-ins with client-cluster methods in a similar way as done in IFCA (Ghosh et al., 2020), CFL (Sattler et al., 2020), FL+HC (Briggs et al., 2020), and so on. Then the server aggregates the plug-ins in the same clusters to keep a controllable number of plug-ins, like in $O(1)$ or $O(log\mathcal{M})$, where $\mathcal{M}$ denotes the number of clients. The server can significantly reduce the number of plug-ins in this way, thus increase the scalability of our method. A simpler method is enough for solving the issue of the massive plug-ins: as our selection method doesn't require the presence of plug-ins, clients may not upload the plug-ins after training. Instead, the server can request the appropriate plug-in from the corresponding client after calculating the selection scores. Considering the common issue in cross-client setting, FL systems may encounter client dropout (Li et al., 2020a; Kairouz et al., 2021; Tan et al., 2022b). In a situation where the client with the most appropriate plug-in is out of connection, the server may attempt to request plug-ins one by one with the selection score. To avoid downloading all plug-ins and training features to clients, if the number of clients grows to a large number, clients can choose to add noise to their local test features and send the noised test features to the server to select the plug-ins. In this way, each client can get the exact plug-in they need without the need to download all the plug-ins and the noised training features, which will cause a great communication cost with a great number of clients in the FL. We conduct experiments to test the feasibility of this method

against the communication and storage burden of HPFL in FL systems with plenty of clients. The result are shown in Table 12.

Table 12: Experiment results of sharing noised test feature. §: we focus more on GFL setting. Numbers in **ForestGreen** highlight highest values in GFL setting. $E_p$ denotes the epoch of fine-tuning. Other hyper-paramters follows the experiments in Table 2.

| Clients | 10 (sample 50% each round) | | | | 100 (5% each round) | | | |
|---|---|---|---|---|---|---|---|---|
| Non-IID | Dir(0.1) | | Dir(0.05) | | Dir(0.1) | | Dir(0.05) | |
| Test Set | GFL§ | PFL | GFL§ | PFL | GFL§ | PFL | GFL§ | PFL |
| Method/Model | GM PM | PM | GM PM | PM | GM PM | PM | GM PM | PM |
| CIFAR-10 | | | | | | | | |
| HPFL($\hat{h}_{test}$) $E_p = 1$ | 81.5 95.4 | 95.4 | 62.4 96.0 | 96.0 | 73.6 **91.7** | 94.9 | 47.9 **85.2** | 93.9 |
| HPFL($\hat{h}_{test}$) $E_p = 10$ | 81.5 **95.7** | 95.7 | 62.4 **96.3** | 96.3 | 73.6 90.3 | 95.7 | 47.9 **85.2** | 95.3 |
| FMNIST | | | | | | | | |
| HPFL($\hat{h}_{test}$) $E_p = 1$ | 86.0 98.3 | 98.3 | 76.1 99.1 | 99.1 | 90.2 97.9 | 97.9 | 86.1 **95.3** | 98.1 |
| HPFL($\hat{h}_{test}$) $E_p = 10$ | 86.0 **98.4** | 98.4 | 76.1 **99.2** | 99.2 | 90.2 **98.6** | 98.8 | 86.1 94.0 | 98.7 |
| CIFAR-100 | | | | | | | | |
| HPFL($\hat{h}_{test}$) $E_p = 1$ | 68.6 **75.7** | 83.3 | 65.3 **78.8** | 87.4 | 59.7 **67.5** | 81.2 | 47.9 72.7 | 84.1 |
| HPFL($\hat{h}_{test}$) $E_p = 10$ | 68.6 69.5 | 85.7 | 65.3 78.0 | 88.9 | 59.7 63.8 | 84.1 | 47.9 **75.5** | 86.4 |
| Tiny-ImageNet-200 | | | | | | | | |
| HPFL($\hat{h}_{test}$) $E_p = 1$ | 56.5 51.8 | 70.8 | 54.9 55.5 | 74.7 | 47.2 **58.6** | 71.3 | 42.1 **59.2** | 74.7 |
| HPFL($\hat{h}_{test}$) $E_p = 10$ | 56.5 47.4 | 73.7 | 54.9 50.3 | 77.0 | 47.2 57.7 | 73.2 | 42.1 57.8 | 76.5 |

### F.2.2 ONLINE INFERENCE OF HPFL

Our method is designed to infer in batches, online test-time adaptation where test samples arrive one by one is not our main application scenario (Hoi et al., 2021; Jiang & Lin, 2022; Tan et al., 2023). Selecting plug-ins for every sample may incur expensive computation costs. However, when met with a similar situation and the computation costs are unavoidable, clients can give up downloading plug-ins. Similar to method stated in Appendix F.2.1, following instructions free clients from the storage burden brought by downloading plug-ins and corresponding training features for every sample: simply send the noised test feature to the server; let the server select the appropriate plug-in; then infer at the server side; and finally return the inference result back to the client. In this way, communication and latency issues with online inference can be solved with slight modifications in our proposed method.

