# OpenReview forum: "All for One and One for All: A Collaborative FL Framework for Generic Federated Learning with Personalized Plug-ins"
_ICLR.cc/2024/Conference — Submitted to ICLR 2024_

### Official Review · Reviewer_3cLo · 2023-10-28

**Soundness:** 2 fair
**Presentation:** 2 fair
**Contribution:** 2 fair
**Rating:** 5
**Confidence:** 4

**Summary:**

The authors propose a federated learning scheme that captures the advantages of both generic FL and personalized FL. The feature extractor can be trained based on any generic FL strategy (e.g., FedAvg). Then, the remaining plug-in module $\rho$ is trained based on any personalized FL method. This part is then uploaded to the server. During test-time, each client can selectively download the plug-in module and make the prediction. Experiments show that the methodology performs well on both generic and personalized datasets.

**Strengths:**

1. Taking advantage of both generic FL and personalized FL is interesting and important.

2. Experimental results are promising.

3. The paper is generally easy to follow.

**Weaknesses:**

1. I believe the test-time strategy of this method is somewhat impractical and unclear:
- Whenever a test sample appears, the scheme requires the client to download all the features from the server (which is same as the number of clients in the system), and compute (8). This can incur communication, computation, and latency issues.
- Moreover, each client may need to download multiple plug-in modules during inference, which will again incur additional communication and latency. For example, suppose there are three clients, each having classes (1, 2), (3, 4), (5, 6) in its local dataset, and produce plug-in modules 1, 2, 3, respectively. Now for an arbitrary client X, when the first test sample belongs to class 1, client X will download the plug-in module 1. Then, if the next test sample belongs to class 2, client X will download the plug-in module 2. Finally, it will also download plug-in module 3 whenever classes 5 or 6 appears in the test set. Is this really happening in the proposed method?
- Finally, it is not clear how the server should generate $h_m$ tailored to a specific client. For example, again consider an example with three clients having classes (1, 2), (3, 4), (5, 6). If class 1 appears as a test sample in client 1, what should the server send to this client? In other words, what are $h_1$, $h_2$, $h_3$ in this case?

2. The discussion with the prior work (Chen & Chao, 2021) is not clear to me. In (Chen & Chao, 2021), the authors are also designing a generic feature extractor, and two plug-in heads: one is used for the generic scenario and the second is used for the personalized scenario. If the authors aim to do both generic and personalized testing during inference, one may simply choose between the generic plug-in module and the personalized plug-in module developed in (Chen & Chao, 2021), instead of selecting the plug-in modules from all clients in the system (this can significantly improve resource consumptions while achieving the similar performance). What is the advantage/contribution of the authors work compared to these methods/concepts? I'm not sure whether the authors' approach will provide any advantages compared to this method. The authors should provide comprehensive experiments as well as discussions regarding this. Related to this comment, probably the authors' scheme can have advantage compared to (Chen & Chao, 2021) when considering mixed dataset of personalized and generic dataset similar to as done in [Ref]: Each client's test dataset has more local classes when the distribution shift is small, while having more classes outside of the client's local dataset when the distribution shift is large.

(Chen & Chao, 2021) ON BRIDGING GENERIC AND PERSONALIZED
FEDERATED LEARNING FOR IMAGE CLASSIFICATION

[Ref] SplitGP: Achieving both generalization and personalization in federated learning, arXiv e-prints (2022): arXiv-2212.

**Questions:**

Please refer to the weakness above.

---

> ### Author Response · Authors · 2023-11-16
> **Response to Reviewer 3cLo - Part 1**
>
> We truly thank the reviewer for taking the time to review. We appreciate that you find our proposed problem interesting and important, the result of our method promising, and the paper is easy to follow. According to your valuable comments, we provide detailed feedback. Due to similar concerns of different reviewers on the scalability and privacy safety of our method, which are all solvable by uploading the noise test features to the server for selection plug-ins, we have conducted experiments to testify the feasibility of this slight modification against communication and storage burden of HPFL in FL systems with plenty of clients. The results are in the [Solution toward shared scalability and privacy concern](https://openreview.net/forum?id=8rhHI6C8iC&noteId=ULBhGHenqN). Please find special responses to your comments below.
>
> **Q1.1: scalability problem**
> >Whenever a test sample appears, the scheme requires the client to download all the features from the server (which is same as the number of clients in the system), and compute (8). This can incur communication, computation, and latency issues.
>
> **Authors' Response:** Thanks for pointing out potential problems when HPFL meets with an FL system with a great number of clients. As the number of clients grows to a large number, clients can choose to add noise to their local test features and send the noised test features to the server to select the plug-ins. In this way, each client can get the required plug-in without the need to download all the plug-ins and the noised training features, which cause a great communication cost with a great number of clients in the FL. We conduct experiments to testify the feasibility of this method against the communication burden of HPFL in FL systems with plenty of clients. Please find them in the [Solution toward shared scalability and privacy concern](https://openreview.net/forum?id=8rhHI6C8iC&noteId=ULBhGHenqN).
>
> **Q1.2: Online test-time adaption**
> >Moreover, each client may need to download multiple plug-in modules during inference, which will again incur additional communication and latency. For example, suppose there are three clients, each having classes (1, 2), (3, 4), (5, 6) in its local dataset, and produce plug-in modules 1, 2, 3, respectively. Now for an arbitrary client X, when the first test sample belongs to class 1, client X will download the plug-in module 1. Then, if the next test sample belongs to class 2, client X will download the plug-in module 2. Finally, it will also download plug-in module 3 whenever classes 5 or 6 appears in the test set. Is this really happening in the proposed method?
>
> **Authors' Response:** HPFL is designed to infer in batches, online inference where test samples arrive one by one is not our main application scenario [1, 2, 3]. Selecting plug-ins for every sample may incur expensive computation costs. However, when met with a similar situation and the computation costs are unavoidable, clients can give up downloading plug-ins. Similar to method stated in [Solution toward shared scalability and privacy concern](https://openreview.net/forum?id=8rhHI6C8iC&noteId=ULBhGHenqN), following instructions free clients from the storage burden brought by downloading plug-ins and corresponding training features for every sample: simply send the noised test feature to the server; the server select the appropriate plug-in; **then infer at the server side; and finally return the inference result back to the client**. In this way, communication and latency issues with online inference can be solved with slight modifications in HPFL. We have added ***Appendix F.2.2 in our revision*** to discuss how HPFL can adjust to special circumstances like online inference.
>
> **Q1.3: clarification on details of our method**
> >Finally, it is not clear how the server should generate $h_m$ tailored to a specific client. For example, again consider an example with three clients having classes (1, 2), (3, 4), (5, 6). If class 1 appears as a test sample in client 1, what should the server send to this client? In other words, what are $h_1$, $h_2$, $h_3$ in this case?
>
> **Authors' Response:** We appreciate your effort to understand our method in detail. However, $h_m$ is not tailored to a specific client during the inference period, and it is uploaded by clients with the training data used to finetune plug-ins. In the case mentioned by the reviewer, $h1, h2, h3$ will respectively be the feature of training data in client 1,2,3 and have nothing to do with test samples. Suppose $\xi_m \sim \mathcal{D}_m$ is local training data sampled from the local distribution of client m, $h_m=g(\xi_m)$ where $g$ is the global shared feature extractor. In other words, $h_m$ is the feature of client m's local training data, obtained with a single forward pass through the global feature extractor. We will be grateful if you can point out the presentation in our paper that mislead you to the misunderstanding of $h_m$.

---

> ### Author Response · Authors · 2023-11-16
> **Response to Reviewer 3cLo - Part 2**
>
> **Q2: comparison with FedRoD**
> >The discussion with the prior work (Chen & Chao, 2021) is not clear to me. In (Chen & Chao, 2021), the authors are also designing a generic feature extractor, and two plug-in heads: one is used for the generic scenario and the second is used for the personalized scenario. If the authors aim to do both generic and personalized testing during inference, one may simply choose between the generic plug-in module and the personalized plug-in module developed in (Chen & Chao, 2021), instead of selecting the plug-in modules from all clients in the system (this can significantly improve resource consumptions while achieving the similar performance). **(a) What is the advantage/contribution of the authors work compared to these methods/concepts?** I'm not sure whether the authors' approach will provide any advantages compared to this method. **(b)The authors should provide comprehensive experiments as well as discussions regarding this.** Related to this comment, **\(c\) probably the authors' scheme can have advantage compared to (Chen & Chao, 2021) when considering mixed dataset of personalized and generic dataset similar to as done in [7]**: Each client's test dataset has more local classes when the distribution shift is small, while having more classes outside of the client's local dataset when the distribution shift is large.
>
> **Authors' Response:** **(a)** While FedRoD [4] can handle both GFL-GM and PFL-PM by two plug-in heads, our proposed setting GFL-PM is different from GFL-GM [5] and PFL-PM [6], because in our setting distribution of local test data may not the same as that of training data, in this situation, a single personalized classifier is not enough for the diversified test data. **(b)** Our experiment also shows that FedRoD's local classifiers trained on local training data can't handle test data in different distributions, leading to poor performance in our proposed GFL-PM setting. It fails to provide a more fine-grained solution compared with our methods. As shown in ***Table 2 of origin main text***, even using the global classifier to infer on every client in the GFL-PM setting (it is the same as the performance in the GFL-GM setting as the model in every client is identical), our method shows a significant performance gain compared with FedRoD in most cases. **\(c\)** Our proposed set-up GFL-PM is totally different from the setting in [7], which is an interpolated dataset of local test data and global test data. In our set-up, clients may meet with local test data from both other clients and themselves. Since the problem we are trying to solve is quite divergent, we are not sure if our proposed method HPFL is able to perform well in setting in [7], as we assign a single plug-in from other clients or the client itself to the client, instead of interpolating the local head and the global head.
>
> >  ***Reference***
> >
> >  [1] Hoi, Steven CH, Doyen Sahoo, Jing Lu, and Peilin Zhao. "Online learning: A comprehensive survey." Neurocomputing, 2021.
> >
> >  [2] Jiang, Liangze, and Tao Lin. "Test-Time Robust Personalization for Federated Learning." In ICLR, 2022.
> >
> >  [3] Tan, Yue, Chen Chen, Weiming Zhuang, Xin Dong, Lingjuan Lyu, and Guodong Long. "Is Heterogeneity Notorious? Taming Heterogeneity to Handle Test-Time Shift in Federated Learning." In NeruIPS, 2023.
> >
> >  [4] Chen, Hong-You, and Wei-Lun Chao. "On Bridging Generic and Personalized Federated Learning for Image Classification." In ICLR, 2021.
> >
> >  [5] Communication-efficient learning of deep networks from decentralized data. In AISTATS, 2017.
> >
> >  [6] Arivazhagan, Manoj Ghuhan, Vinay Aggarwal, Aaditya Kumar Singh, and Sunav Choudhary. "Federated learning with personalization layers." arXiv, 2019.
> >
> >  [7] SplitGP: Achieving both generalization and personalization in federated learning, arXiv, 2022.

---

> ### Author Response · Authors · 2023-11-20
> **Welcome for more discussions from Reviewer 3cLo**
>
> Dear reviewer 3cLo,
>
> We appreciate your time for reviewing and helpful comments, according to which we devote maximum effort to addressing the concerns and enriching the discussions in our paper accordingly. We **summarize our response** as follows:
>
> - (1.1) **Scalability problem**: To address the potential problems when HPFL meets with an FL system with a great number of clients, we propose a simple but effective scheme to **choose plug-ins with the noised test features**, and test its performance to the availability of this scheme enhancing scalability.
> - (1.2) **Online test-time adaption**: We claim that **original purpose of HPFL is inferring in batches**; however, HPFL also provides a solution in online inference settings with several modifications in the process of HPFL with **inference on the server**, without **the communication and latency issues**.
> - (1.3) **Clarification on details of $h_m$**: In response to the reviewer's confusion about the details of training data features $h_m$, we explain the definition of $h_m$ and how to get $h_m$ in detail.
> - (2) **Comparison with FedRoD**: According to the reviewer's questions about the discussion with FedRoD, we highlight the difference between HPFL and FedRoD as **limited adaptation to test data of FedRoD's local classifiers** , which lead to poor performance of FedRoD in GFL-PM. In response to the reviewer's suggestion, we also analyze the difference between our proposed GFL-PM setting and the setting mentioned by the reviewer.
>
> Hopefully, your questions have been well addressed with our responses. If you have any additional concerns or comments that we may have missed in our responses, we would be most grateful for any further feedback from you to help us further enhance our work.
>
> Best regards
>
> Authors of #2503

---

> ### Comment · Reviewer_3cLo · 2023-11-21
> **Response to authors**
>
> Thanks for the authors for providing the responses. I still have concerns on the scalability issue and the practicability of the algorithm:
>
> 1. Regarding the solution toward scalability, do you mean that clients should send the features of all test samples to the server? This will cause significant communication burden at the clients. Note that other baselines do not require any communication during inference. The privacy guarantee is also not sure even when the noise is added. A more detailed discussion is needed in terms of differential privacy if the authors choose this way of inference.
>
> 2. Regarding the on-line inference, if the authors are making inference per batch, are the accuracies getting averaged over multiple batches when reporting the results? I also have a concern regarding this batch-wise inference: Again, suppose there are three clients, each having classes (1, 2), (3, 4), (5, 6) in its local dataset, and produce plug-in modules 1, 2, 3, respectively. Now suppose that a test batch of a specific client has classes 1, 3, 5. Which modules should the client download? All modules will cause some problems in this case. How can the current solution handle this issue when making inference in batch-wise?

---

> > ### Author Response · Authors · 2023-11-21
> > **Thanks for swift reply and further explanation for Reviewer 3cLo**
> >
> > Dear reviewer 3cLo,
> >
> > We appreciate your swift response and further questions despite such a busy period, according to which we further explain our solution toward scalability and scheme for inference.
> >
> > > **(a) Regarding the solution toward scalability, do you mean that clients should send the features of all test samples to the server? This will cause significant communication burden at the clients.** Note that other baselines do not require any communication during inference. **(b) The privacy guarantee is also not sure even when the noise is added. A more detailed discussion is needed in terms of differential privacy if the authors choose this way of inference.**
> >
> > **Further Response:** **(a)** Yes, actually, due to the relatively low dimension of features, communication of test features tends not to be a heavy burden for clients. Take ResNet-18 used in our main experiment as an example; the number of classifier's feature dimensions is 512, while communicating the whole model needs to transmit around 11 million parameters; the communication burden of features is far less than the model even with thousands of test samples in a single client in cross-device settings, where a massive number of clients exist and a client usually possess relatively few data compared with those in cross-silo settings. **(b)** In cross-silo settings, there is no need to upload test features; we can simply follow the paradigm of the original HPFL with significantly fewer clients in the FL systems. We are sorry if there is a potential misleading in our previous response; differential privacy is used to protect potential privacy issues of backbone training using GFL algorithms like FedAvg [1, 2] rather than inference. As for the shared noised features, we discuss and testify the privacy safety of sharing the noised features as in HPFL in ***Appendix E.2***.
> >
> > > Regarding the on-line inference, if the authors are making inference per batch, are the accuracies getting averaged over multiple batches when reporting the results? I also have a concern regarding this batch-wise inference: Again, suppose there are three clients, each having classes (1, 2), (3, 4), (5, 6) in its local dataset, and produce plug-in modules 1, 2, 3, respectively. Now suppose that a test batch of a specific client has classes 1, 3, 5. Which modules should the client download? All modules will cause some problems in this case. How can the current solution handle this issue when making inference in batch-wise?
> >
> > **Further Response:** We think our proposed setting GM-PFL is more practical as demonstrated in the ***3rd paragraph in the introduction of main text*** and ***real-world scenarios in Appendix F.1***, where clients encounter only one distribution at a time instead of the mixture of multiple test distributions at the same time. In the real-world scenario, maybe it is more common that a client sequentially meets test batches with classes (1, 2), (3, 4), (5, 6) instead of (1, 3, 5). However, we agree the reviewer's opinion that meeting classes like (1, 3, 5) is a universal setting, where HPFL can only provide more options for the clients instead of promising to well handle this situation, as we state in the ***Response to Q4 of the reviewer ENEM***: if there is really no similar data pattern even in the whole FL system, HPFL can only attempt to choose the most appropriate plug-in, though in this situation, even if the most appropriate plug-in may not perform well on the totally strange test data due to not identical training and test distribution. It is also worth noting that with the increasing number of clients in the FL system, this situation happens significantly less with the abundance of clients' training data distribution.
> >
> > We sincerely hope our responses address your further questions well. If you still have any additional concerns or comments, please don't hesitate to discuss them with us. We would be most grateful for any further feedback from you to help us improve our work.
> >
> > >  **Reference**
> > >
> > >  [1] Abadi, Martin, Andy Chu, Ian Goodfellow, H. Brendan McMahan, Ilya Mironov, Kunal Talwar, and Li Zhang. "Deep learning with differential privacy." In Proceedings of the 2016 ACM SIGSAC conference on computer and communications security, 2016.
> > >
> > >  [2] Geyer, Robin C., Tassilo Klein, and Moin Nabi. "Differentially private federated learning: A client level perspective." arXiv, 2017.
> >
> > Best regards
> >
> > Authors of #2503

---

> ### Comment · Reviewer_3cLo · 2023-11-21
> **Thanks**
>
> Thanks for the clarification. I'm still not convinced especially with the second response. The authors' main focus is on data distributional difference between training and testing. It is very natural to have a new data distribution during testing that is different from the clients' training data distribution in the system. In cross-silo setting, this will be a huge problem. Even in cross-device setting, I don't think simply saying that there will be no issues due to a large number of clients is not enough. There could still be new clients when the task is complicated. I will keep my original score.

---

> > ### Author Response · Authors · 2023-11-22
> > **Thanks for response from Reviewer 3cLo**
> >
> > Dear reviewer 3cLo,
> >
> > We really appreciate your swift responses despite such a busy period. We admitted that HPFL cannot well handle the situation the reviewer proposes in the previous response, but compared with classic GFL and PFL methods, HPFL provides more choices for the clients to choose from. Moreover, Our main focus is on the situation where a client meets with test data in a similar distribution of other clients' training data instead of a broader definition which the reviewer may regard as a totally unseen distribution in the whole FL system, as stated in both the ***3rd paragraph in the introduction of main text*** and clarification of our proposed setting **GFL-PM in Section 3.3**.
> >
> > We sincerely hope our responses make the aim of our paper more clear to you. And thanks again for your time spent reviewing our work.
> >
> > Best regards
> >
> > Authors of #2503

---

### Official Review · Reviewer_TW8h · 2023-10-30

**Soundness:** 2 fair
**Presentation:** 3 good
**Contribution:** 2 fair
**Rating:** 5
**Confidence:** 4

**Summary:**

This paper introduces Hot-Pluggable Federated Learning (HPFL), a framework that addresses the challenge of adapting federated learning (FL) models to test data distributions via combining a shared global feature extractor and selected personalized plug-in modules based on test data. During inference, a selection algorithm helps clients choose suitable plug-in modules to achieve high generalization. Experimental results show that HPFL outperforms other FL algorithms and has potential applications in some FL scenarios, such as continual federated learning and one-shot FL.

**Strengths:**

- The paper's framework clearly illustrates the process and main modules of the proposed framework, making the method easy to understand.
- The idea of applying HPFL to continual federated learning, one-shot federated learning, and a federated learning plugin market is interesting, and the authors provide a figure of the effectiveness of continual federated learning with HPFL.
- The relevant findings in the experiments regarding the impact of plugins at different layers on performance are interesting and insightful.

**Weaknesses:**

- The combination of using a separate backbone and classifier in PFL for improvement, along with the idea of customize the classifier on local client, is not novel. The former has been widely validated for its effectiveness in FedPer and FedRep, while the latter's idea is highly relevant to that of KNN-Per [1] and FedBABU [2]. However, the authors do not provide relevant discussion or experimental comparisons of the latter ones.
- Since the authors compared the performance of both GFL and PFL, the experiments should include more commonly used GFL-related baselines.
- There is a missing detailed analysis of potential privacy risks in uploading and sharing local features.


[1] Marfoq, Othmane, et al. "Personalized federated learning through local memorization." International Conference on Machine Learning. PMLR, 2022.

[2] Oh, Jaehoon, SangMook Kim, and Se-Young Yun. "FedBABU: Toward Enhanced Representation for Federated Image Classification." International Conference on Learning Representations. 2021.

**Questions:**

Regarding the experimental data, I have some concerns.
- The performance of PerFedMask (Table 1) in the personalized setting is much lower than other baselines. I wonder if there was an error in implementing this method, or if its hyper-parameters were not appropriately fine-tuned?
- About the experiments with added noise (Table 4). Why do experiments with different levels of noise produce nearly the same results? This appears somewhat counterintuitive. Can the authors provide a more detailed analysis?

---

> ### Author Response · Authors · 2023-11-16
> **Response to Reviewer TW8h - Part 1**
>
> We would like to express our gratitude for the time the reviewer spent reviewing. We appreciate that you find our method easy to follow, the application of our method interesting, and the finding in the experiments insightful. According to your valuable comments, we provide detailed feedback.
>
> **Q1: comparison with KNN-Per FedBABU**
> >The combination of using a separate backbone and classifier in PFL for improvement, along with the idea of customize the classifier on local client, is not novel. The former has been widely validated for its effectiveness in FedPer and FedRep, while the latter's idea is highly relevant to that of **KNN-Per (a)** [1] and **FedBABU (b)** [2]. However, the authors do not provide relevant discussion or experimental comparisons of the latter ones.
>
> **Authors' Response:** Thanks for providing us with two excellent methods in the PFL setting for comparing our methods with. **KNN-Per (a)** [1] and **FedBABU (b)** [2] share the similar problem of classic PFL algorithms [3-6] in our GFL-PM setting.
> - **KNN-Per (a)** build a local datastore $S_m$ with local training data and query $S_m$ to assist inference using a knn method. Imagine a circumstance where some classes in local test data are not presented in local training data, in this situation, the test samples of those classes are likely to retrieve a wrong label from $S_m$ and mislead the final decision.
> - In **FedBABU (b)**, classifiers of local clients are randomly initialized and never updated during federated training, and for personalization during the evaluation process, FedBABU fine-tuned its classifiers on local training data. Here, FedBABU also requires local test data to be in the same distribution as local training data to fine-tune the classifiers on local training data to perform well on local test data.
>
> **These two methods only customize the classifier to fit static local test data with the same distribution as its training data.** Instead, our proposed setting GFL-PM is faced with the problem of changing local test data distributions, whose distribution is different from local training data, and HPFL is proposed to solve this problem. These two methods don't propose **the adaption mechanism to test data, which is implemented with plug-in selection in HPFL**, thus probably face with similar difficulty of classic PFL algorithms like FedPer and FedRep when being exploited in GFL-PM setting.
>
> **Q2: GFL baseline**
> >Since the authors compared the performance of both GFL and PFL, the experiments should include more commonly used GFL-related baselines.
>
> **Authors' Response:** We'd like to thank you for pointing out the potential deficiency in GFL baselines. For comparison in the GFL setting in our experiments, FedAvg is a classic and most commonly used GFL algorithm, and FedRod can simultaneously achieve leading generic and personalized performance. For more comprehensive comparison in the GFL setting, we prepare to compare HPFL with a more recent SOTA GFL method FedSAM [7]. However, due to the time needed to carry out the experiments, we may need several days to report the result of FedSAM and add the result to our revision.
>
> **Q3: privacy risk of sharing the noised features**
> >There is a missing detailed analysis of potential privacy risks in uploading and sharing local features.
>
> **Authors' Response:** Due to the page limit, in ***Appendix E.2***, we discussed potential privacy risks and conducted experiments of model inversion attack [8] to testify the privacy safety of sharing noisy features obtained with our method. As the experiments show, after adding noise in a way as stated in ***Section 4.3 of origin main text***, even an attacker with the global model can hardly recover the original image, while with raw feature and the model, an attacker can easily recover the raw image, which shows the effectiveness of our privacy protection method. We also discuss other possible methods to protect privacy, like averaging the raw features [9] or choosing plug-ins with methods without the requirements of sharing direct information about raw data or features like OOD detection [10]. To increase the visibility of our discussions about privacy in ***Appendix E.2***, we have directed the reader to them in ***Section 4.3 of main text in our revision***.

---

> ### Author Response · Authors · 2023-11-16
> **Response to Reviewer TW8h - Part 2**
>
> **Q4: experimental results of PerFedMask**
> >The performance of PerFedMask (***Table 1 of origin main text***) in the personalized setting is much lower than other baselines. I wonder if there was an error in implementing this method, or if its hyper-parameters were not appropriately fine-tuned?
>
> **Authors' Response:** Our results of PerFedMask is presented in ***Table 2 of origin main text*** instead of ***Table 1 of origin main text***. We followed the open-source implementation of PerFedMask [6]. To fairly compare it with other baselines and HPFL, we only change the data partition method, the learning rate scheduler, and the communication round to align with other methods. To testify the scalability of methods, we increase the number of clients, and raise batch size from 50 to 128 as the value we use in other methods for better convergence. All other hyperparameters, including the initial learning rate, stay the same as the origin implementation of PerFedMask. We have added these information about hyperparameters to ***Appendix C.4 in our revision***.
>
> **Q5: experiments with different levels of noise**
> >About the experiments with added noise (***Table 4 of origin main text***). Why do experiments with different levels of noise produce nearly the same results? This appears somewhat counterintuitive. Can the authors provide a more detailed analysis?
>
> **Authors' Response:** For this counterintuitive phenomenon, we provide our explaination in ***Section 5.3 of origin main text***. Since our selection method only chooses plug-ins with selection scores, if the changes in selection scores are not significant enough to affect the final choices of plug-ins, which can be observed in the MMD scores heatmaps in ***Figure 3 of origin main text***, the performance of HPFL will remain the same, which adds robustness towards noise to HPFL without significant loss on accuracy performance. We have added a detailed discussion about this phenomenon in ***Section 5.3 of our revision***.
>
> >  ***Reference***
> >
> > [1] Marfoq, Othmane, et al. "Personalized federated learning through local memorization." In ICML, 2022.
> >
> > [2] Oh, Jaehoon, SangMook Kim, and Se-Young Yun. "FedBABU: Toward Enhanced Representation for Federated Image Classification." In ICLR, 2021.
> >
> > [3] Collins, Liam, Hamed Hassani, Aryan Mokhtari, and Sanjay Shakkottai. "Exploiting shared representations for personalized federated learning." In ICML, 2021.
> >
> > [4] Arivazhagan, Manoj Ghuhan, Vinay Aggarwal, Aaditya Kumar Singh, and Sunav Choudhary. "Federated learning with personalization layers." arXiv, 2019.
> >
> > [5] Liang, Paul Pu, Terrance Liu, Liu Ziyin, Nicholas B. Allen, Randy P. Auerbach, David Brent, Ruslan Salakhutdinov, and Louis-Philippe Morency. "Think locally, act globally: Federated learning with local and global representations." arXiv, 2020.
> >
> > [6] Setayesh, Mehdi, Xiaoxiao Li, and Vincent WS Wong. "PerFedMask: Personalized Federated Learning with Optimized Masking Vectors." In ICLR, 2022.
> >
> > [7] Qu, Zhe, Xingyu Li, Rui Duan, Yao Liu, Bo Tang, and Zhuo Lu. "Generalized federated learning via sharpness aware minimization." In ICML, 2022.
> >
> > [8] Nanxuan Zhao, Zhirong Wu, Rynson WH Lau, and Stephen Lin. What makes instance discrimination good for transfer learning? In ICLR, 2020.
> >
> > [9] Luo, Mi, Fei Chen, Dapeng Hu, Yifan Zhang, Jian Liang, and Jiashi Feng. "No fear of heterogeneity: Classifier calibration for federated learning with non-iid data." In NeruIPS, 2021.
> >
> > [10] Ren, Jie, Peter J. Liu, Emily Fertig, Jasper Snoek, Ryan Poplin, Mark Depristo, Joshua Dillon, and Balaji Lakshminarayanan. "Likelihood ratios for out-of-distribution detection." In NeruIPS, 2019.

---

> ### Author Response · Authors · 2023-11-20
> **Welcome for more discussions from Reviewer TW8h**
>
> Dear reviewer TW8h,
>
> Thanks for reviewing and constructive comments, from which we put forth our best effort to answer the questions and enrich the content of our paper accordingly. A **summary of our responses** for your convenience is as follows:
>
> - (1) **comparison with KNN-Per & FedBABU**: To answer your question on the difference of HPFL compared with KNN-Per and FedBABU, we review the **main steps of both KNN-Per and FedBABU**, and analyze **their limitation when met with GFL-PM**. Then we conclude that the **lack of the adaption mechanism to test data** put obstacles in applying KNN-Per and FedBABU in GFL-PM setting.
> - (2) **GFL baseline**: In response to the reviewer's advice to add more GFL baselines, we point out the existing GFL baselines FedAvg and FedRoD, and compare HPFL with a more recent SOTA GFL method **FedSAM**.
> - (3) **Privacy risk of sharing the noised features**: In response to the reviewer's concerns about sharing the noised features, we directed the reviewer to **Appendix E.2**, where we discussed potential privacy risks and privacy protection methods, and conducted **experiments of model inversion attack** using the shared features and the global backbone in HPFL.
> - (4) **Experimental results of PerFedMask**: We explain **the missing details of PerFedMask** in our responses and add these details in **Appendix C.4 in our revision**.
> - (5) **Experiments with different levels of noise**: We direct the reviewer to the explanation of the counterintuitive phenomenon that experiments with different levels of noise produce nearly the same results in **Section 5.3 of origin main text**, and further discuss the possible cause of this phenomenon in our responses.
>
> Hopefully, your questions have been well addressed with our responses. If you have any additional concerns or comments that we may have missed in our responses, we would be most grateful for any further feedback from you to help us further enhance our work.
>
> Best regards
>
> Authors of #2503

---

> > ### Comment · Reviewer_TW8h · 2023-11-22
> >
> > The author's explanations and the addition of the baseline effectively addressed my concerns about the problem setting and the lack of comparative methods. Additionally, the author's inclusion of model inversion attacks provided a deeper discussion of the privacy risks associated with feature sharing. I am pleased to raise my score.
> >
> > However, I still have concerns about the novelty of this framework. Test-time adaptation in the FL setting is not a new setting, and although I am interested in the potential applications of this framework (such as continual learning, one-shot FL, and the FL plugin marketplace), the author has only presented very preliminary experimental results on continual learning. Furthermore, the method's novelty is still limited; compared to kNN-Per, its main improvement lies in leveraging shared features from other clients during inference.

---

> > ### Author Response · Authors · 2023-11-22
> > **Thanks for your swift reply and raising the score**
> >
> > Dear Reviewer TW8h,
> >
> > We're grateful for your timely feedback during this busy period. We deeply appreciate your consideration in raising the score.  Your constructive comments have significantly contributed to the refinement of our work.
> >
> > **Novelty of our framework:** Our framework has some differences from test-time adaptation in FL: (i) previous test-time adaptation in FL, like in FedTHE [1] assumes the distribution shift comes from independent change of local data distribution and has nothing to do with other clients, like in FedTHE; to the best of our knowledge, we are the **first to propose and formulate the setting where clients meet test data in the similar/same distribution of other clients' training data**; (ii) instead of directly adapting to the new distribution of test data, we reduce the origin GFL-GM setting which is more challenging to solve into a more easy-to-solve problem SFL, and adapt to the new test distribution in this problem framework to get better adaption performance compared with previous test-time adaptation methods such as FedTHE in our proposed setting GFL-PM, which is shown in ***Table 2 of main text***.
> >
> > **Potential applications:** We appreciate the reviewer's interest in the potential applications of our framework; to better testify the potential of our framework, we will add more detailed discussion and demonstrate more results in our revision.
> >
> > **Novelty of our method:** Due to the generality of HPFL, we tend to use **simple methods for GFL backbone training, PFL plug-in fine-tuning and plug-in selection**, for the purpose of keeping the demonstration of HPFL as simple as possible. Actually, HPFL can combine with knn-per with further designed selection method and practical method to ensure privacy and scalability; we think it may be achieved by uploading subsets of/whole datastores as in knn-per under proper privacy protection; comparing similarities of datastores and test features of the client and select proper datastores; then downloading the specific datastore and carrying out inference with knn method. Therefore, our paper only provides a basic paradigm for HPFL, and **HPFL can have a significantly different form from knn-per if backbone training, plug-in fine-tuning, plug-in selection with more sophisticated methods**.
> >
> > Thank you again for your prompt response during this busy period and support for our work. We remain open and ready to dive into any more questions or suggestions you might have.
> >
> > >  ***Reference***
> > >
> > >  [1] Jiang, Liangze, and Tao Lin. "Test-Time Robust Personalization for Federated Learning." In ICLR, 2022.
> >
> >
> > Best regards and thanks,
> >
> > Authors of #2503

---

> ### Author Response · Authors · 2023-11-22
> **Result of new GFL baseline FedSAM**
>
> We show the results of the new GFL baseline **FedSAM** in ***Table 3*** below, and compare FedSAM and HPFL, as suggested in the reviewer's comments. Unless otherwise stated, the setting remains the same as the main experiments in Table 2 of origin main text, exclusive hyper-parameters for FedSAM is specified in ***Appendix C.4***:
>
> **Table 3**
>
> | Clients | 10 |  |  |  |  |  | 100 |  |  |  |  |  |
> |:---------:|:----:|:----:|:----:|:----:|----:|:----:|:-----:|:-----:|:-----:|:-----:|:-----:|:-----:|
> | Non-IID | Dir(0.1) |  |  | Dir(0.05) |  |  | Dir(0.1) |  |  | Dir(0.05) |  |  |
> | Test Set | $\text{GFL}^{\S}$ | $\text{GFL}^{\S}$ | PFL | $\text{GFL}^{\S}$ | $\text{GFL}^{\S}$ | PFL | $\text{GFL}^{\S}$ | $\text{GFL}^{\S}$ | PFL | $\text{GFL}^{\S}$ | $\text{GFL}^{\S}$ | PFL |
> | Method/Model | GM | PM | PM | GM | PM | PM | GM | PM | PM | GM | PM | PM |
> | **CIFAR-10** | | | | | | | | | | | | |
> | FedSAM $E_p=1$ | 84.5 | 47.8 | 96.0 | 65.4 | 33.2 | 96.6 | 50.4 | 36.6 | 88.5 | 36.6 | 23.3 | 90.1 |
> | FedSAM $E_p=10$ | 84.5 | 42.3 | 96.0 | 65.4 | 28.8 | 96.7 | 50.4 | 25.4 | 90.3 | 36.6 | 15.4 | 91.3 |
> | HPFL $E_{p}=1$ | 81.5 | 95.4 | 95.4 | 62.4 | 96.0 | 96.0 | 73.6 | **88.6** | 94.9 | 47.9 | **82.2** | 93.9 |
> | HPFL $E_{p}=10$ | 81.5 | **95.7** | 95.7 | 62.4 | **96.3** | 96.3 | 73.6 | 85.7 | 95.7 | 47.9 | 81.8 | 95.3 |
> | **FMNIST** | | | | | | | | | | | | |
> | FedSAM $E_p=1$ | 89.3 | 53.8 | 98.5 | 77.2 | 36.6 | 99.4 | 86.3 | 76.9 | 98.2 | 85.2 | 70.8 | 98.3 |
> | FedSAM $E_p=10$ | 89.3 | 51.7 | 98.5 | 77.2 | 33.9 | 99.4 | 86.3 | 64.2 | 98.5 | 85.2 | 52.2 | 98.6 |
> | HPFL(MMD) $E_{p}=1$ | 86.0 | 98.3 | 98.3 | 76.1 | 99.0 | 99.1 | 90.2 | 97.6 | 97.9 | **86.1** | 81.4 | 98.1 |
> | HPFL(MMD) $E_{p}=10$ | 86.0 | **98.4** | 98.4 | 76.1 | **99.1** | 99.2 | 90.2 | **97.9** | 98.8 | **86.1** | 74.1 | 98.7 |
> | **CIFAR-100** | | | | | | | | | | | | |
> | FedSAM $E_p=1$ | 68.4 | 57.4 | 84.9 | 64.1 | 43.0 | 88.0 | 41.3 | 27.3 | 67.5 | 34.8 | 18.4 | 73.8 |
> | FedSAM $E_p=10$ | 68.4 | 46.0 | 85.6 | 64.1 | 30.7 | 88.8 | 41.3 | 18.6 | 71.1 | 34.8 | 11.6 | 77.3 |
> | HPFL(MMD) $E_{p}=1$ | 68.6 | **74.8** | 83.3 | 65.3 | **75.8** | 87.4 | 59.7 | **63.8** | 81.2 | 47.9 | **72.3** | 84.1 |
> | HPFL(MMD) $E_{p}=10$ | 68.6 | 72.2 | 85.7 | 65.3 | 73.9 | 88.8 | 59.7 | 55.7 | 84.1 | 47.9 | 70.9 | 86.4 |
> | **Tiny-ImageNet-200** | | | | | | | | | | | | |
> | FedSAM $E_p=1$ | 57.0 | 48.6 | 73.3 | 55.0 | 42.1 | 77.1 | 43.8 | 30.4 | 66.8 | 38.0 | 12.7 | 72.0 |
> | FedSAM $E_p=10$ | 57.0 | 41.0 | 75.1 | 55.0 | 32.1 | 78.2 | 43.8 | 21.0 | 69.3 | 38.0 | 21.1 | 72.0 |
> | HPFL(MMD) $E_{p}=1$ | 56.5 | **51.9** | 70.8 | 54.9 | 58.5 | 74.7 | 47.2 | **50.7** | 71.3 | 42.1 | **47.1** | 74.7 |
> | HPFL(MMD) $E_{p}=10$ | 56.5 | 50.9 | 73.7 | 54.9 | **58.8** | 77.0 | 47.2 | 48.0 | 73.2 | 42.1 | 43.9 | 76.5|
>
> The **bold** numbers mean they are the best GFL results compared with other methods in Table 3. From the results, we observe that though FedSAM achieves excellent GFL-GM performance in some settings, HPFL outperforms FedSAM in all GFL settings in the experiment.

---

### Official Review · Reviewer_ENEM · 2023-10-31

**Soundness:** 3 good
**Presentation:** 2 fair
**Contribution:** 3 good
**Rating:** 6
**Confidence:** 4

**Summary:**

This paper studies a relatively new and general setup of personalized FL that serves several clients possibly from a new distribution and/or holding only unlabeled test data.

The framework is based on general FL with several customized plug-in modules learned by personalized FL. Initially, clients acquire a global shared feature extractor. Subsequently, utilizing this feature extractor as a fixed foundation, clients proceed to train multiple personalized plug-in modules individually. These personalized modules are tailored to the specifics of their local data and are then stored in a modular repository on the server.

During the inference stage, a selection algorithm is employed that enables clients to make precise choices, selecting and downloading the most appropriate plug-in modules from the modular repository. The experiments show that the method achieves high generalization performance, particularly when confronted with target data distributions that may differ significantly from the initial training data.

**Strengths:**

* The paper proposes an interesting aspect extending the existing literature of bridging generic FL and personalized FL. The proposed selective FL framework is sound in some FL scenarios.
* The paper provides discussions and solutions on the practical drawbacks of selective FL including privacy concerns and the system overhead.
* Extensions to other cases like continual learning are appreciated
* Limitations, impact, and related works are properly discussed.

**Weaknesses:**

* Some parts of the design choices of the proposed selective FL could be elaborated more to give the paper a broader scope. For example, the part of the general feature extractors can be trained with any GFL algorithm. Considering there are many GFL methods proposed, what are their performances in comparison? Are they compatible with selective FL?  For the selection methods in section 4.3, could the author provide more insights and discussion besides the used distance-based methods? Any other potential replacements?

* Considering FedRoD is the sota in this topic evidenced by Table 2, can the authors provide a more fine-grained comparison? Specifically, FedRod (also followed by FedTHE) proposes to train the general model with a class-balanced loss; will this improve HPFL as well? FedRoD also fine-tunes the whole general model for each client with local training; how many personalized epochs are trained like the reported HPFL (e.g., E_p=1 or 10)?

* There are many typos and inconsistent wording

**Questions:**

Can the authors provide a discussion on the data/client assumptions on when should HPFL work or not work? Suppose the new client has very different distributions unseen in the training, how possible will HPFL work well?

---

> ### Author Response · Authors · 2023-11-16
> **Response to Reviewer ENEM - Part 1**
>
> We genuinely appreciate the reviewer's efforts. We are delighted to know that you find our proposed framework sound, the extension of our method interesting, and the discussions and solutions on the practical drawbacks make sense. According to your valuable comments, we provide detailed feedback.
>
> **Q1: Ablation study on backbone training and selection methods**
> >Some parts of the design choices of the proposed selective FL could be elaborated more to give the paper a broader scope. For example, the part of the general feature extractors can be trained with any GFL algorithm. **(a) Considering there are many GFL methods proposed, what are their performances in comparison? Are they compatible with selective FL?** **(b) For the selection methods in section 4.3, could the author provide more insights and discussion besides the used distance-based methods? Any other potential replacements?**
>
> **Authors' Response:** Thanks for the advice on improving the ablation study of our method. **(a)** As long as the used GFL methods are able to train a strong general feature extractor, HPFL is able to utilize the feature extractor to train the personalized plug-ins and extract features. Due to the great availability of GFL methods used for training backbone [1-5], we didn't carry out experiments to see how different GFL backbone training methods affect the overall performance of HPFL. We add experiments using FedRoD [6] to testify HPFL's compatibility with other GFL methods. The results are given in ***Table 2*** below. Number of clients equal $10$, local fine-tuning epoch $E_p=10$, local datasets are partitioned in $Dir(0.1)$. Other settings remain the same as the main experiments in ***Table 2 of origin main text***:
>
> **Table 2**
>
> |                   |                  | HPFL (FedAvg) | HPFL (FedRoD) |
> | :-----------------: | :----------------: | :-------------: | :-------------: |
> | **CIFAR-10**      | GFL-GM           | 81.5          | 85.3(+3.8)     |
> |                   | GFL-PM           | 95.7          | 96.0(+0.3)     |
> |                   | PFL-PM           | 95.7          | 96.0(+0.3)     |
> | **FMNIST**        | GFL-GM           | 86.0          | 87.9(+1.9)     |
> |                   | GFL-PM           | 98.4          | 98.4(+0)       |
> |                   | PFL-PM           | 98.4          | 98.4(+0)       |
> | **CIFAR-100**     | GFL-GM           | 68.6          | 69.9(+1.3)     |
> |                   | GFL-PM           | 72.2          | 68.5(-3.7)     |
> |                   | PFL-PM           | 85.7          | 85.5(-0.2)     |
> | **Tiny-ImageNet-200** | GFL-GM        | 56.5          | 57.4(+0.9)     |
> |                   | GFL-PM           | 50.9          | 56.0(+5.1)     |
> |                   | PFL-PM           | 73.7          | 74.7(+1.0)     |
>
>
> From the overall performance of HPFL(FedRoD), we can see that HPFL using FedRoD as its backbone training method is comparable to that using FedAvg, which confirms HPFL is compatible with the GFL methods other than FedAvg. Following your valuable comments, in ***Section 5.2 of main text of our revision***, we have directed readers to results of HPFL(FedRoD) shown in ***Appendix D.6 in our revision*** to discuss the ablation study of GFL backbone training methods. **(b)** As for selection methods other than distance-based methods, due to the page limit, we choose to introduce other selection methods in the main body. In the ***Appendix D.2***, we elaborately discuss an OOD-Detection (Out of Distribution Detection)-based method [7] to select suitable plug-ins by judging whether the test data is OOD for plug-ins and choose the plug-in with the lowest OOD score. Aside from OOD-Detection-based methods, if a method can be used to judge how appropriate plug-ins are for certain test data, it is available for selection. Even a supervised classifier trained with the noised training features as data and corresponding client indexes as labels may be able to select plug-ins.

---

> ### Author Response · Authors · 2023-11-16
> **Response to Reviewer ENEM - Part 2**
>
> **Q2: experimental result of FedRod**
> >Considering FedRoD is the sota in this topic evidenced by ***Table 2 of origin main text***, can the authors provide a more fine-grained comparison? **(a) Specifically, FedRod (also followed by FedTHE) proposes to train the general model with a class-balanced loss; will this improve HPFL as well?** FedRoD also fine-tunes the whole general model for each client with local training; **(b) how many personalized epochs are trained like the reported HPFL (e.g., E_p=1 or 10)**?
>
> **Authors' Response:** Yes, and thank the reviewer for the suggestions about a more elaborated discussion on FedRoD. **(a)** From the results of HPFL(FedRoD) given in ***Response to Q1***, we observe an interesting fact that even if FedRoD shows excellent performance in GFL-GM (surpasses FedAvg in many settings), fine-tuning the backbone trained with it is not advantageous as shown in PFL-PM (only comparable with fine-tuning on the backbone trained with FedAvg). From this phenomenon, we presume the advantage of FedRoD in GFL-GM should mainly be attributed to its global head trained with a class-balanced loss instead of its backbone. **(b)** As is stated in ***Section 5.1 of origin main text***, the local training epoch of all methods is $1$. To avoid the possible misunderstanding, we would like to clarify that the local training epoch is different from the $E_p$ in HPFL, because the backbone will not be updated when fine-tuning the plug-ins in HPFL. Fine-tuning of HPFL happens after the GFL backbone training and is completely done in local clients. We apologize for not clearly indicating personalized epochs of other baseline methods with personalized epochs like PerFedMask. We have specified the value of personalized epochs for those methods with personalized epochs in ***Table 2 of main text in our revision***.
>
>
> **Q3: presentation**
> >There are many typos and inconsistent wording
>
> **Authors' Response:** We apologize for the typos and inconsistent wording. We will check the whole paper thoroughly and carefully, and revise mistakes in it.
>
> **Q4: How HPFL tackles very different distributions unseen in the training**
> >**(a) Can the authors provide a discussion on the data/client assumptions on when should HPFL work or not work?** **(b) Suppose the new client has very different distributions unseen in the training, how possible will HPFL work well?**
>
> **Authors' Response:** Sure. **(a)** HPFL works well in situations where the distribution of local training data and test data is quite different, but its test data distribution is similar to the distribution of other clients' training data, HPFL can help the client by borrowing knowledge from other clients, i.e., utilize plug-ins trained by other clients. **(b)** For the clients that meet with totally different test data distributions that haven't been seen in the training data of all clients (include those clients themselves; if those clients have similar distribution on training data and test data, they can choose the plug-in trained on their own local data). However, if there is really no similar data pattern even in the whole FL system, HPFL can only attempt to choose the most appropriate plug-in, though in this situation, even if the most appropriate plug-in may not perform well on the totally strange test data due to not identical training and test distribution [8]. But it is worth noting that with the increasing number of clients in the FL system, this situation happens significantly less with the abundance of clients' training data distribution.
>
> >  ***Reference***
> >
> >  [1] Communication-efficient learning of deep networks from decentralized data. In AISTATS, 2017.
> >
> >  [2] Li, Tian, Anit Kumar Sahu, Manzil Zaheer, Maziar Sanjabi, Ameet Talwalkar, and Virginia Smith. "Federated optimization in heterogeneous networks." In MLSys, 2020.
> >
> >  [3] Wang, Jianyu, Qinghua Liu, Hao Liang, Gauri Joshi, and H. Vincent Poor. "Tackling the objective inconsistency problem in heterogeneous federated optimization." In NeurIPS, 2020.
> >
> >  [4] Karimireddy, Sai Praneeth, Satyen Kale, Mehryar Mohri, Sashank Reddi, Sebastian Stich, and Ananda Theertha Suresh. "Scaffold: Stochastic controlled averaging for federated learning." In ICML, 2020.
> >
> >  [5] Durmus, Alp Emre, Zhao Yue, Matas Ramon, Mattina Matthew, Whatmough Paul, and Saligrama Venkatesh. "Federated Learning Based on Dynamic Regularization." In ICLR, 2021.
> >
> >  [6] Chen, Hong-You, and Wei-Lun Chao. "On Bridging Generic and Personalized Federated Learning for Image Classification." In ICLR, 2021.
> >
> >  [7] Ren, Jie, Peter J. Liu, Emily Fertig, Jasper Snoek, Ryan Poplin, Mark Depristo, Joshua Dillon, and Balaji Lakshminarayanan. "Likelihood ratios for out-of-distribution detection." In NeruIPS, 2019.
> >
> >  [8] Quinonero-Candela, Joaquin, Masashi Sugiyama, Anton Schwaighofer, and Neil D. Lawrence, eds. Dataset shift in machine learning. Mit Press, 2008.

---

> ### Author Response · Authors · 2023-11-20
> **Welcome for more discussions from Reviewer ENEM**
>
> Dear reviewer ENEM,
>
> Thanks for reviewing and constructive comments, from which we put forth our best effort to answer the questions and enrich the content of our paper accordingly. A **summary of our responses** for your convenience is followed:
>
> - (1) **Ablation study on backbone training and selection methods**: According to the appreciated suggestion of the reviewer, to testify the compatiablity of HPFL, we discuss the possibility of HPFL backbone training with FedRoD and provide the **result of HPFL(FedRoD)**. As for selection methods, we direct the reviewer to the **Appendix D.2 of main text** for other selection methods and introduce **a simple supervised selection method**.
> - (2) **Experimental result of FedRod**: We direct the reviewer to the discussion about **HPFL using FedRoD for backbone training** and avoid possible misunderstanding by clarifying training details of FedRoD.
> - (3) **Presentation**: In response to the reviewer's concerns on paper presentation, we have checked the whole paper thoroughly and carefully, and revise its mistakes in our revision.
> - (4) **HPFL's intuition dealing unseen distributions**: To answer the possible confusion of the reviewer on HPFL's ability to **unseen distributions in local training data**, we dicuss HPFL under different circumstances: (i) HPFL when test data distribution is similar to the distribution of **other clients’ training data**; (ii) and HPFL with **totally different test data distributions** that haven’t been seen in the training data of all clients (include those clients themselves).
>
> We hope your questions have been well addressed with our responses. If you have any additional concerns or comments that we may have missed in our responses, we would be most grateful for any further feedback from you to help us further enhance our work.
>
> Best regards
>
> Authors of #2503

---

> ### Author Response · Authors · 2023-11-22
> **Window for responsing and draft updating is closing after 20 hours**
>
> Dear Reviewer ENEM,
>
>
> Thanks very much for your time and valuable comments. We understand you’re busy. But as the window for responsing and paper revision is closing, would you mind checking our response (a brief summary, and details) and confirm whether you have any further questions? We are very glad to provide answers and revision to your further questions.
>
>
> Best regards and thanks,
>
> Authors of #2503

---

> ### Author Response · Authors · 2023-11-23
> **Would you mind raising the score**
>
> Dear reviewer ENEM,
>
> Thanks for your great efforts in reviewing our paper and your support for our work. We appreciate that you find our proposed problem interesting; with discussions and solutions on the practical drawbacks of naive solution for selective FL; interesting and appreciated applications; and proper discussion of limitations, impact, and related works.
>
> Your valuable comments have greatly helped us improve the presentation of our paper, the integrity of the ablation study, and the clarity of HPFL’s intuition. It is greatly appreciated that the reviewer provide insightful comments about the backbone trained with different GFL methods.
>
> If you have no further questions/concerns, would you mind raising the score?
>
> Best regards and thanks,
>
> Authors of #2503

---

### Official Review · Reviewer_nQmW · 2023-10-31

**Soundness:** 3 good
**Presentation:** 1 poor
**Contribution:** 2 fair
**Rating:** 3
**Confidence:** 4

**Summary:**

The paper motivates a new federated set-up in which, at test time, a client may encounter data drawn from the distribution of other clients in the collaboration. The paper then proposes a solution for this set-up, which proceeds in two stages: (1) learning a global feature extractor, and (2) learning personalized plug-in modules that are stored in the server and selected dynamically at inference time. The paper presents results on a variety of standard datasets with a limited number of clients.

**Strengths:**

- To my knowledge, the introduced paradigm (Equation 3) is novel and the approach to solving it (Theorem 3.2) seems intuitive. I did not check the math carefully.
- The gains over the presented baselines do seem significant. However, presenting error bars would strengthen the claims in the manuscript.
- In the same vein, the evaluation framework presented by the paper, focusing on the global data distribution while also presenting the performance on the local distribution, seems appropriate for the task they present.

**Weaknesses:**

- The main weakness of the paper is its presentation. In particular, the manuscript is not explicit about the form of the test distribution that the clients will encounter. I was confused about this, and could only infer it until I reached Section 3.3 and I finally understood the problem's objective.
- Similarly, it is not clear to me whether the method is proposed for cross-device or cross-silo settings. Although the motivating example in Section 1 is cross-device, the experiments are performed with a limited number of clients (<= 100). More concerning, in the proposed algorithm, the server stores all the training clients' plug-ins. In cross-device scenarios with a massive number of clients, this would not scale. There is no discussion of this in the paper.
- Storing all the clients' plug-ins may also be a privacy risk, as there is not any aggregation that protects the clients' privacy from a malicious server. The threat model is not discussed by the paper.

**Questions:**

- Although the given set-up is novel, I am still not convinced of its importance, at least not justify the drawbacks of the proposed solution. I recommend the manuscript update its introduction to include more motivating examples that are explicitly related to their problem formulation.
- Although the presented evaluation framework is good for the presented problem, a more exhaustive framework could be formulated to rigorously test the approach, e.g., one in which train client $m$ is given test data $n$, and we check if it indeed downloads plug-in $n$.

---

> ### Author Response · Authors · 2023-11-16
> **Response to Reviewer nQmW - Part 1**
>
> We are grateful for the time the reviewer spent on reviewing. It is pleasant to learn that you find our proposed setting novel, our solution intuitive, the performance gains significant, and the evaluation method appropriate. According to your valuable comments, we provide detailed feedback. Due to similar concerns of different reviewers on the scalability and privacy safety of our method, which are all solvable by uploading the noise test features to the server for selection plug-ins, we have conducted experiments to testify the feasibility of this slight modification against communication and storage burden of HPFL in FL systems with plenty of clients. The results are in the [Solution toward shared scalability and privacy concern](https://openreview.net/forum?id=8rhHI6C8iC&noteId=ULBhGHenqN). Please find special responses to your comments below.
>
> **Q1: presentation of paper**
> >The main weakness of the paper is its presentation. In particular, the manuscript is not explicit about the form of the test distribution that the clients will encounter. I was confused about this, and could only infer it until I reached ***Section 3.3 of origin main text*** and I finally understood the problem's objective.
>
> **Authors' Response:** We are sorry that we failed to highlight the test scenarios of our problem, which is briefly illustrated in the ***3rd paragraph in the introduction of origin main text (Section 1)***: "i.e., clients equally meet with the test data from all clients." We chose to elaborate more in our theory part for this problem setting. We have added a detailed explanation, "In GFL-PM setting, the test set every client encounters comes from the local test data of other clients, whose distribution is the same as that of clients' local training data." in the ***3rd paragraph in the introduction of our revision*** to help readers to more easily understand our proposed problem.
>
> **Q2: scalability problem**
> >Similarly, it is not clear to me whether the method is proposed for cross-device or cross-silo setting. Although the motivating example in ***Section 1 of origin main text*** is cross-device, the experiments are performed with a limited number of clients (<= 100). More concerning, in the proposed algorithm, the server stores all the training clients' plug-ins. In cross-device scenarios with a massive number of clients, this would not scale. There is no discussion of this in the paper.
>
> **Authors' Response:** Thanks for pointing out the potential scalability problem of HPFL. In fact, HPFL can be applied to both cross-device and cross-silo setting, with a slight modification in cross-device setting where the number of clients is overwhelmingly large. We introduce the methods to enhance the scalability problem of HPFL as follows:
> - **To handle the massive plug-ins needed to be stored in the server**, the server can cluster the plug-ins with client-cluster methods in a similar way as done in IFCA [1], CFL [2], FL+HC [3], and so on. Then the server aggregates the plug-ins in the same clusters to keep a controllable number of plug-ins, like in $O(1)$ or $O(log(number \ of \ clients))$. The server can significantly reduce the number of plug-ins in this way, thus increase the scalability of our method.
> - **A simpler method is enough for solving the issue of the massive plug-ins**: as our selection method doesn't require the presence of plug-ins, clients may not upload the plug-ins after training. Instead, the server can request the appropriate plug-in from the corresponding client after calculating the selection scores. Considering the common issue in cross-client setting, FL systems may encounter client dropout [4, 5, 6]. In a situation where the client with the most appropriate plug-in is out of connection, the server may attempt to request plug-ins one by one with the selection score.
> - **To avoid downloading all plug-ins and training features to clients**, if the number of clients grows to a large number, clients can choose to add noise to their local test features and send the noised test features to the server to select the plug-ins. In this way, each client can get the exact plug-in they need without the need to download all the plug-ins and the noised training features, which will cause a great communication cost with a great number of clients in the FL. We conduct experiments to test the feasibility of this method against the communication and storage burden of HPFL in FL systems with plenty of clients, please find them in the [Solution toward shared scalability and privacy concern](https://openreview.net/forum?id=8rhHI6C8iC&noteId=ULBhGHenqN).
>
> We have added the discussions of those scalability problems of HPFL in the ***Appendix F.2.1 of our revision***.

---

> ### Author Response · Authors · 2023-11-16
> **Response to Reviewer nQmW - Part 2**
>
> **Q3: privacy concern of sharing plug-ins**
> >Storing all the clients' plug-ins may also be a privacy risk, as there is not any aggregation that protects the clients' privacy from a malicious server. The threat model is not discussed by the paper.
>
> **Authors' Response:** Due to page limit, we discussed the potential privacy risk of sharing plug-ins in ***Appendix E.1***.
> - As stated in ***Section 4.3 of origin main text***, **sharing part of models (i.e. plug-ins) is exposed to actually no more privacy risk than sharing the parameter gradients or the complete global model as adopted in classic FL algorithms like FedAvg** [7], which in practice is often protected with regular privacy protect techniques like differential privacy (DP) [8].
> - **As clients in HPFL don't upload the gradient during plug-ins training**, no gradient of models is accessible for the server, so there is no need to worry about gradient inversion attacks [9-12] towards plug-ins.
> - **We also carry out experiments to testify the safety of our method toward the model inversion attack** [13,14,15] when with both the noise features and the model trained on raw features in ***Appendix E.2***.
> - **Even extreme concern on potential privacy risk of keeping plug-ins in the server can be solved by only requesting plug-ins after selection** as described in ***Response to Q2***, clients providing the plug-ins can ask the server to delete the plug-ins after sending the plug-ins to the clients in need. To increase the visibility of our discussions about the privacy problem of sharing plug-ins in ***Appendix E.1***, we have directed the reader to them in ***Section 4.2 of main text in our revision***.
>
> **Q4: the importance of proposed set-up**
> >Although the given set-up is novel, I am still not convinced of its importance, at least not justify the drawbacks of the proposed solution. I recommend the manuscript update its introduction to include more motivating examples that are explicitly related to their problem formulation.
>
> **Authors' Response:** We appreciate the suggestion given by the reviewer to justify the motivation of our proposed set-up. We give some examples exhibiting the significance of our proposed set-up below:
>
> - **Case 1: Some clients may have insufficient computing resources or local training data to fine-tune a deep learning model in a cross-device setting.** In these situations, training distribution can be regarded as an empty set $\emptyset$. In this way, the client cannot get a personalized model by locally fine-tuning the global model. In traditional GFL and PFL settings, the client has no choice but to adopt the global model and endure the lack of personalization. This problem is caused by the mismatch of training data distribution and test data distribution, as assumed in our proposed set-up, and is solvable with our proposed method HPFL by exploiting personalized plug-ins from other clients.
> - **Case 2: A car with a personalized automated driving system (ADS) has driven out of the previous city it used to be.** It requires to personalize on geometric data from the present city it is now in for improving the performance of the ADS in this new city. Classic GFL and PFL in this situation leave the ADS no option but to collect the geometric data and personalize on it after the collection completes, and accept the temporary performance loss using the previous personalized model before finishing the new personalization, since the distribution of test data has greatly changed. It's another example where the discrepancy between training data (geometric data from the previous city) and test data (geometric data from the present city) threatens the availability of FL systems. While with our proposed method designed to solve the problem, the ADS can attempt to access the plug-ins from car owners living in the present city.
> - **Case 3: Imagine a person is traveling from a high latitude area to an equatorial region, and the recommender system on their phone is supported by federated learning.** If the recommender system uses the personalized model trained when in the high latitude area, it will continue to prompt thick down jackets for the person, which is clearly an unexpected and unreasonable recommendation. With our method, one can get the same recommendation as the local people with plug-ins on their phones without time to fine-tune the model again.
>
> Following your suggestion, we have added the examples listed above to the ***Appendix F.1 of our revision*** due to the page limit.

---

> ### Author Response · Authors · 2023-11-16
> **Response to Reviewer nQmW - Part 3**
>
> **Q5: evaluation framework**
> >Although the presented evaluation framework is good for the presented problem, a more exhaustive framework could be formulated to rigorously test the approach, e.g., one in which train client $m$ is given test data $n$, and we check if it indeed downloads plug-in $n$.
>
> **Authors' Response:** For our proposed methods HPFL, because all clients have the same selection method $C$. So the selection method is only determined by the test data $n$ and not affected by which client carries out the inference as stated in ***Equation 15 in origin main text*** $C_i(\xi_m,i)=argmax_n \space g\left (\xi_n, \xi_m\right)=C(\xi_m)$ in ***Appendix C.3***. In this situation, the evaluation method the reviewer have suggested is formularized as following equation: $\text{Accuracy} \left(\Omega, \theta _1, \ldots, \theta _M\right)=\frac{1}{M} \sum _{i=1}^M \sum _{m=1}^M p _m \mathbb{E} _{\xi _m \sim \mathcal{D} _m} \mathcal{T}\left(f\left(\theta _{C _i(\xi _m,i)}, \xi _m\right), \xi _m\right)$ is the plug-in n downloaded by client m; and the simplified evaluation method used in our implementation is accuracy  of  PM  selected  by  client  i  on  its  own  data: $\text{Accuracy} _{simplified}\left(\Omega, \theta _1, \ldots, \theta _M\right)=\sum _{i=1}^M p _i \mathbb{E} _{\xi _i \sim \mathcal{D} _i}\mathcal{T}(f(\theta _{C(\xi _i)} ^{pfl}, \xi _j), \xi _j)$. We have strictly proven the equivalence of our simplified evaluation framework and the origin evaluation metrics in ***Appendix C.3***.
>
> >  ***Reference***
> >
> >  [1] Ghosh, Avishek, Jichan Chung, Dong Yin, and Kannan Ramchandran. "An efficient framework for clustered federated learning." In NeurIPS, 2020.
> >
> >  [2] Sattler, Felix, Klaus-Robert Müller, and Wojciech Samek. "Clustered federated learning: Model-agnostic distributed multitask optimization under privacy constraints." IEEE transactions on neural networks and learning systems, 2020.
> >
> >  [3] Briggs, Christopher, Zhong Fan, and Peter Andras. "Federated learning with hierarchical clustering of local updates to improve training on non-IID data." In IJCNN, 2020.
> >
> >  [4] Li, Tian, Anit Kumar Sahu, Ameet Talwalkar, and Virginia Smith. "Federated learning: Challenges, methods, and future directions." IEEE signal processing magazine, 2020.
> >
> >  [5] Kairouz, Peter, H. Brendan McMahan, Brendan Avent, Aurélien Bellet, Mehdi Bennis, Arjun Nitin Bhagoji, Kallista Bonawitz et al. "Advances and open problems in federated learning." Foundations and Trends® in Machine Learning 14, 2021.
> >
> >  [6] Tan, Alysa Ziying, Han Yu, Lizhen Cui, and Qiang Yang. "Towards personalized federated learning." IEEE Transactions on Neural Networks and Learning Systems, 2022.
> >
> >  [7] Communication-efficient learning of deep networks from decentralized data. In AISTATS, 2017.
> >
> >  [8] Wei, Kang, Jun Li, Ming Ding, Chuan Ma, Howard H. Yang, Farhad Farokhi, Shi Jin, Tony QS Quek, and H. Vincent Poor. "Federated learning with differential privacy: Algorithms and performance analysis." IEEE Transactions on Information Forensics and Security, 2020.
> >
> >  [9] Ligeng Zhu, Zhijian Liu, and Song Han. "Deep leakage from gradients." In NeurIPS, 2019.
> >
> >  [10] Zhao, Bo, Konda Reddy Mopuri, and Hakan Bilen. "idlg: Improved deep leakage from gradients." arXiv, 2020.
> >
> >  [11] Jonas Geiping, Hartmut Bauermeister, Hannah Dröge, and Michael Moeller. Inverting gradients–how easy is it to break privacy in federated learning? In NeurIPS, 2020.
> >
> >  [12] Yin, Hongxu, Arun Mallya, Arash Vahdat, Jose M. Alvarez, Jan Kautz, and Pavlo Molchanov. "See through gradients: Image batch recovery via gradinversion." In CVPR, 2021.
> >
> >  [13] Hitaj, Briland, Giuseppe Ateniese, and Fernando Perez-Cruz. "Deep models under the GAN: information leakage from collaborative deep learning." In CCS, 2017.
> >
> >  [14] Nanxuan Zhao, Zhirong Wu, Rynson WH Lau, and Stephen Lin. What makes instance discrimination good for transfer learning? In ICLR, 2020.
> >
> >  [15] Lyu, Lingjuan, Han Yu, Xingjun Ma, Chen Chen, Lichao Sun, Jun Zhao, Qiang Yang, and S. Yu Philip. "Privacy and robustness in federated learning: Attacks and defenses." IEEE transactions on neural networks and learning systems, 2022.

---

> ### Author Response · Authors · 2023-11-20
> **Welcome for more discussions from Reviewer nQmW**
>
> Dear reviewer nQmW,
>
> Thanks for sparing your valuable time to review and offer insightful comments, from which we have exerted maximum effort to answer the questions and thoroughly improve our paper. We **summarize our responses** for your convenience:
>
> - (1) **Presentation of paper**: According to the appreciated suggestion of the reviewer, we provide **a more detailed explanation** for the form of the test distribution and revise our paper to improve the readability.
> - (2) **Scalability issues**: We show that slight modifications on implementation boost the scalability of HPFL, which the reviewer kindly reminded us to discuss. We demonstrate the schemes against the scalability issues from several perspectives: (i) a **cluster-based method** against massive plug-ins stored in **the server**; (ii) a **quicker and easier** scheme against the aforementioned issue; (iii) massive plug-ins and training features **downloaded to clients**.
> - (3) **Privacy issues**: In response to the reviewer's concerns on sharing plug-ins, we analyze **the privacy issue of HPFL** and find there are **no more privacy risks than GFL methods** like FedAvg. We also direct the reviewer for **supportive experiment results** on model inversion attack against HPFL. We additionally discuss an approach to accommodate **extreme privacy concerns of sharing plug-ins**.
> - (4) **Importance of proposed set-up**: We list several **real-world examples** to convince the commonness and importance of the proposed set-up GFL-PM, and show **the difficulty classic GFL and PFL algorithm met in those situations**.
> - (5) **evaluation framework**: To better illustrate **the equivalence** of our implementation and the evaluation method the reviewer suggests in HPFL, we list the formulas of both the evaluation method the reviewer suggests and our implementation and show that they are the same **when all clients use the same selection method as in HPFL**.
>
>
> Hopefully, our responses have addressed your concerns well. If you have any additional concerns or comments that we may have missed in our responses, we would be most grateful for any further feedback from you to help us further enhance our work.
>
> Best regards
>
> Authors of #2503

---

> ### Author Response · Authors · 2023-11-22
> **Window for responsing and draft updating is closing after 20 hours**
>
> Dear Reviewer nQmW,
>
> Thanks a lot for your efforts in reviewing paper and your constructive comments. We understand you’re busy. Thanks very much for your valuable comments. We sincerely understand you’re busy. But as the window for responsing and paper revision is closing, would you mind checking our response (a brief summary, and details) and confirm whether you have any further questions? We look forward to answering more questions from you.
>
> Best regards and thanks,
>
> Authors of #2503

---

> > ### Comment · Reviewer_nQmW · 2023-11-22
> > **Response to the authors**
> >
> > Dear authors,
> >
> > Thank you so much for your response. I appreciate the careful comments, and I apologize for the delayed response.
> >
> > I still have concerns about the scalability of the approach, particularly if the method is supposed to work for both cross-device and cross-silo scenarios. The authors mention that the server can cluster the plug-ins, but this is not the presented and evaluated approach. The authors also mention that the server could request the plug-ins from the clients, but this would require the clients to be stateful, which is only feasible in the cross-silo scenario, where the number of plug-ins would be reasonable and scalability may not be an issue.
> >
> > In terms of privacy, I am not an expert in differential privacy, so I can't comment on whether the noise is enough to guarantee privacy. The privacy principle I'm concerned is the data minimization one, which would prompt the method to get rid of unnecessary plug-ins (by, for example, aggregating them). As discussed above, the strategies presented by the authors require a bit more nuance.
> >
> > At this time, I maintain my rating.

---

> > > ### Author Response · Authors · 2023-11-23
> > > **Thanks for the quick reply and further explanation for Reviewer nQmW**
> > >
> > > Dear reviewer nQmW,
> > >
> > > We appreciate your response and further questions despite such a busy period, according to which we further explain our solution toward scalability and scheme for inference.
> > >
> > > > The authors mention that the server can cluster the plug-ins, but this is not the presented and evaluated approach.
> > >
> > > **Further Response:** We are sorry that we cannot provide a draft solution using a similar clustering method as in IFCA [1], CFL [2], FL+HC [3] instead of the results of clustering plug-ins due to the time limit.
> > >
> > > > The authors also mention that the server could request the plug-ins from the clients, but this would require the clients to be stateful.
> > >
> > > **Further Response:** About the reviewer's concern about the feasibility of our scheme sharing the noised test features caused by stateless clients in cross-device settings, our previous response provides a simple method: "Considering the common issue in cross-client setting, FL systems may encounter client dropout. In a situation where the client with the most appropriate plug-in is out of connection, the server may attempt to request plug-ins one by one with the selection score.". If the method is not so effective in the situation where all suitable plug-ins' owners disconnect with the server, we would like to provide another simple way to address the reviewer's concern: If the **similarities among plug-ins** are higher than a threshold, the plug-ins can be aggregated/abandoned and then stored in the server, in this way, HPFL can also be relieved from the storage burden of plug-ins.
> > >
> > > > In terms of privacy, I am not an expert in differential privacy, so I can't comment on whether the noise is enough to guarantee privacy. The privacy principle I'm concerned is the data minimization one, which would prompt the method to get rid of unnecessary plug-ins (by, for example, aggregating them). As discussed above, the strategies presented by the authors require a bit more nuance.
> > >
> > > **Further Response:** Actually, our selection method based on the similarities of the noised features conforms to the privacy principle about data minimization. As stated above, we can **remove unnecessary plug-ins to reduce the privacy risk of sharing plug-ins**. If **similarities among plug-ins** are higher than a threshold, the plug-ins can be aggregated/abandoned. However, we may not be able to provide further results to demonstrate the feasibility of this method since the discussion period is about to close.
> > >
> > > We sincerely hope our responses address your further concerns well. If you still have any additional concerns or comments, please don't hesitate to discuss them with us. We would be most grateful for any further feedback from you to help us improve our work before the closing of the discussion window.
> > >
> > > >  **Reference**
> > > >
> > > >  [1] Ghosh, Avishek, Jichan Chung, Dong Yin, and Kannan Ramchandran. "An efficient framework for clustered federated learning." In NeurIPS, 2020.
> > > >
> > > >  [2] Sattler, Felix, Klaus-Robert Müller, and Wojciech Samek. "Clustered federated learning: Model-agnostic distributed multitask optimization under privacy constraints." IEEE transactions on neural networks and learning systems, 2020.
> > > >
> > > >  [3] Briggs, Christopher, Zhong Fan, and Peter Andras. "Federated learning with hierarchical clustering of local updates to improve training on non-IID data." In IJCNN, 2020.
> > >
> > > Best regards
> > >
> > > Authors of #2503

---

### Official Review · Reviewer_MHGx · 2023-11-06

**Soundness:** 2 fair
**Presentation:** 3 good
**Contribution:** 1 poor
**Rating:** 5
**Confidence:** 5

**Summary:**

The paper proposes an FL framework called HPFL that addresses a problem named selective federated learning, which aims to balance the trade-off between generic federated learning and personalized federated learning in the presence of data heterogeneity and distribution shifts. It consists of a globally trained shared feature extractor and multiple personalized trained plug-in modules. The authors conduct comprehensive experiments to validate the proposed methods.

**Strengths:**

* The paper introduces a new problem, SFL, which bridges the generic and personalized FL methods.
* The paper is well written, and the experiments are comprehensive.

**Weaknesses:**

* The paper has low novelty. Federated learning and fine-tuning the sub-modules are really common and baseline personalized FL methods.

**Questions:**

See weaknesses.

---

> ### Author Response · Authors · 2023-11-16
> **Response to Reviewer MHGx - Part 1**
>
> We sincerely thank the reviewer for taking the time to review. We appreciate that you find our proposed problem novel, our paper well-written, and the experiments comprehensive. Based on your valuable comments, we provide detailed feedbacks as follows.
>
> **Question: novelty**
> >The paper has low novelty. Federated learning and fine-tuning the sub-modules are really common and baseline personalized FL methods.
>
> We not only propose a novel problem setting but also introduce a novel mechanism to solve this problem. We propose a novel problem named GFL-PM and reduce the problem into Selective FL (SFL), and finally solve the SFL problem with our proposed method HPFL, which introduces a novel plug-in mechanism. Our difference with other previous personalized FL methods, which carry out GFL and fine-tuning the sub-modules, is listed below:
> |            |                       Personalize part                       |                     Personalized method                      | Ability to adapt to test data | Ability to utilize personalized knowledge from other clients |
> | :--------: | :----------------------------------------------------------: | :----------------------------------------------------------: | :---------------------------: | :----------------------------------------------------------: |
> |   FedPer [1]   |                         last layers                          |                         fine-tuning                          |             **×**             |                            **×**                             |
> | LG-FedAVG [2]  |                       previous layers                        |                  only aggregate last layers                  |             **×**             |                            **×**                             |
> |   FedRep [3]   |                          last layer                          |                only aggregate previous layers                |             **×**             |                            **×**                             |
> |   FedRoD [4]  |                          last layer                          |                only aggregate previous layers                |             **×**             |                            **×**                             |
> | PerFedMask [6] |                 determine by calculated mask                 |                         fine-tuning                          |             **×**             |                            **×**                             |
> |   FedBN [7]   |                          BN layers                           |               only aggregate layers except BNs               |             **×**             |                            **×**                             |
> |   FedTHE [5]  |                          last layer                          |                         fine-tuning                          |             **√**             |                            **×**                             |
> |    HPFL    | not limit (experiments are carried out with last layers, however, HPFL is compatible with all kinds of PFL) | not limit (experiments are carried out with fine-tuning, however, HPFL is compatible with all kinds of PFL) |             **√**             |                            **√**                             |
>
> From the table above, we can clearly see most PFL algorithms lack the ability to adapt to test data. FedTHE utilizes an interpolation of global prediction and personalized prediction, whose coefficient is trained on test data. While our proposed method HPFL obtains similar ability with selection method and plug-ins from other clients, which enables the exploitation of personalized knowledge from other clients simultaneously. Therefore, the novelty of HPFL does not lie in how the backbone is trained or how the plug-ins are personalized, which is the key point of most previous methods, but in its overall framework and plug-ins selection mechanism.

---

> ### Author Response · Authors · 2023-11-16
> **Response to Reviewer MHGx - Part 2**
>
> >  ***Reference***
> >
> >  [1] Arivazhagan, Manoj Ghuhan, Vinay Aggarwal, Aaditya Kumar Singh, and Sunav Choudhary. "Federated learning with personalization layers." arXiv, 2019.
> >
> >  [2] Liang, Paul Pu, Terrance Liu, Liu Ziyin, Nicholas B. Allen, Randy P. Auerbach, David Brent, Ruslan Salakhutdinov, and Louis-Philippe Morency. "Think locally, act globally: Federated learning with local and global representations." arXiv, 2020.
> >
> >  [3] Collins, Liam, Hamed Hassani, Aryan Mokhtari, and Sanjay Shakkottai. "Exploiting shared representations for personalized federated learning." In ICML, 2021.
> >
> >  [4] Chen, Hong-You, and Wei-Lun Chao. "On Bridging Generic and Personalized Federated Learning for Image Classification." In ICLR, 2021.
> >
> >  [5] Jiang, Liangze, and Tao Lin. "Test-Time Robust Personalization for Federated Learning." In ICLR, 2022.
> >
> >  [6] Setayesh, Mehdi, Xiaoxiao Li, and Vincent WS Wong. "PerFedMask: Personalized Federated Learning with Optimized Masking Vectors." In ICLR, 2022.
> >
> >  [7] Li, Xiaoxiao, Meirui JIANG, Xiaofei Zhang, Michael Kamp, and Qi Dou. "FedBN: Federated Learning on Non-IID Features via Local Batch Normalization." In ICLR, 2020.

---

> ### Author Response · Authors · 2023-11-19
> **Welcome for more discussions from Reviewer MHGx**
>
> Dear reviewer MHGx,
>
> Thanks for your time in reviewing our paper. According to your comments, we have tried our best to convince the reviewer of the novelty of our paper. Here is a **summary of our response** for your convenience:
>
> - **Novelty issues**: We clarify the novelty of our proposed problem setting **GFL-PM**; reduction of **GFL-PM** into **Selective FL**; and solution to the SFL problem HPFL, with a novel **plug-in** mechanism. To further explain the novelty of our proposed method HPFL, We provide a table comparing different methods with our proposed method HPFL, demonstrating that the ability to adapt to test data and share personalized knowledge in an FL system are the key differences between HPFL and those classic PFL algorithms.
>
> We humbly hope our response has addressed your concerns about the novelty of our paper. If you have any additional concerns or comments that we may have missed in our responses, we would be most grateful for any further feedback from you to help us further enhance our work.
>
> Best regards
>
> Authors of #2503

---

> ### Author Response · Authors · 2023-11-22
> **Window for responsing and draft updating is closing after 20 hours**
>
> Dear Reviewer MHGx,
>
> Thanks a lot for your time in reviewing, and we appreciated your precious suggestions for improving our work. We sincerely understand you’re busy. But as the window for responsing and paper revision is closing, would you mind checking our response (a brief summary, and details) and confirm whether you have any further questions? We look forward to answering more questions from you.
>
> Best regards and thanks,
>
> Authors of #2503

---

> ### Author Response · Authors · 2023-11-23
> **Would you mind raising the score**
>
> Dear reviewer MHGx,
>
> Thanks for your great efforts in reviewing our paper. We appreciate that you find our proposed problem novel and interesting, our paper is well-written with comprehensive experimental results.
>
> Your constructive comments have greatly helped us improve the statement of our novelty. Especially, the suggestion of providing a more detailed comparation of personalized FL methods with federated learning and fine-tuning the sub-modules is critical.
>
> If you have no further questions/concerns, would you mind raising the score?
>
> Best regards and thanks,
>
> Authors of #2503

---

### Author Response · Authors · 2023-11-16
**Solution toward shared scalability and privacy concern**

Due to similar concerns of different reviewers on the scalability and privacy safety of our method, which are all solvable by uploading the noise test features to the server for selection plug-ins, we have conducted experiments to testify the feasibility of this slight modification against communication and storage burden of HPFL in FL systems with plenty of clients. The new results are given in ***Table 1-1 and 1-2*** below. Accuracies in each cell are arranged in the order of GFL-GM, GFL-PM, and PFL-PM. Unless otherwise stated, the setting remains the same as the main experiments in ***Table 2 of origin main text***:

**Table 1-1**

**10 client**
|                   |          | $\alpha$=0.1 |          |        | $\alpha$=0.05 |          |        |
| :-----------------: | :------: | :----------: | :------: | :----: | :-----------: | :------: | :----: |
|                   |          |    GFL-GM    |  GFL-PM  | PFL-PM |    GFL-GM     |  GFL-PM  | PFL-PM |
| CIFAR-10          | $E_p$=1  |     81.5     |   95.4   |  95.4  |     62.4      |   96.0   |  96.0  |
|                   | $E_p$=10 |     81.5     | **95.7** |  95.7  |     62.4      | **96.3** |  96.3  |
| FMNIST            | $E_p$=1  |     86.0     |   98.3   |  98.3  |     76.1      |   99.1   |  99.1  |
|                   | $E_p$=10 |     86.0     | **98.4** |  98.4  |     76.1      | **99.2** |  99.2  |
| CIFAR-100         | $E_p$=1  |     68.6     | **75.7** |  83.3  |     65.3      | **78.8** |  87.4  |
|                   | $E_p$=10 |     68.6     |   69.5   |  85.7  |     65.3      |   78.0   |  88.9  |
| Tiny-ImageNet-200 | $E_p$=1  |     56.5     |   51.8   |  70.8  |     54.9      |   55.5   |  74.7  |
|                   | $E_p$=10 |     56.5     |   47.4   |  73.7  |     54.9      |   50.3   |  77.0  |

**Table 1-2**

**100 client**
|                   |        | $\alpha$=0.1 |          |        | $\alpha$=0.05 |          |        |
| :-----------------: | :----: | :----------: | :------: | :----: | :-----------: | :------: | :----: |
|                   |        |    GFL-GM    |  GFL-PM  | PFL-PM |    GFL-GM     |  GFL-PM  | PFL-PM |
| CIFAR-10          | $E_p$=1  |     73.6     | **91.7** |  94.9  |     47.9      | **85.2** |  93.9  |
|                   | $E_p$=10 |     73.6     |   90.3   |  95.7  |     47.9      | **85.2** |  95.3  |
| FMNIST            | $E_p$=1  |     90.2     |   97.9   |  97.9  |     86.1      | **95.3** |  98.1  |
|                   | $E_p$=10 |     90.2     | **98.6** |  98.8  |     86.1      |   94.0   |  98.7  |
| CIFAR-100         | $E_p$=1  |     59.7     | **67.5** |  81.2  |     47.9      |   72.7   |  84.1  |
|                   | $E_p$=10 |     59.7     |   63.8   |  84.1  |     47.9      | **75.5** |  86.4  |
| Tiny-ImageNet-200 | $E_p$=1  |     47.2     | **58.6** |  71.3  |     42.1      | **59.2** |  74.7  |
|                   | $E_p$=10 |     47.2     |   57.7   |  73.2  |     42.1      |   57.8   |  76.5  |

The bold numbers mean they are the best GFL results compared with methods in **Table 2 of origin main text (including HPFL with no noises on test features)**. From the results, we observe that adding noise to test features doesn't hurt or even boost the performance of HPFL. We think the reason behind the phenomenon is that the noise added to test and training data cancel each other out to some extent, but we don't theoretically analyze this effect. With this experiment, we find sending the noise test features to the server for selection plug-ins is practical to relieve clients from the heavy communication and storage burden encountered when the number of clients rapidly increases.

---

### Meta-Review · Area_Chair_RAGo · 2023-12-12

**Metareview:**

This paper has been assessed by five knowledgeable reviewers. Four of them recommended rejecting it (one straight reject score, three marginal rejects), while one opted for marginal acceptance. The paper introduces a new method for bridging generic and personalized Federated Learning. The reviewers would like to see a more detailed than provided analysis of potential privacy risks. They also brought up some serious scalability challenges that would likely hinder practical usability of the proposed method in either cross-silo or cross-device settings, and raised concerns about the novelty of the presented framework. As a result of authors engaging the reviewers in a discussion, one of them had raised their score. However, all things considered, this promising work is not yet ready to be including in the program of ICLR 2024.

**Justification For Why Not Higher Score:**

Reviewers have raised serious concerns and only some of them had been clarified via discussion. This work may have legs but it is not ready for prime time yet.

**Justification For Why Not Lower Score:**

n/a

---

### Decision · Program_Chairs · 2024-01-16

Reject